# MS-IMAP: A Multi-Scale Graph Embedding Approach for Interpretable Manifold Learning

**Shay Deutsch**[*][§]                          *shaydeu1@gmail.com*

**Lionel Yelibi**[†][§]                          *ylblio001@myuct.ac.za*

**Alex Tong Lin**[†][§]                          *atlin271@gmail.com*

**Arjun Ravi Kannan**[§]                          *arjun.kannan@gmail.com*

**Reviewed on OpenReview:** *https://openreview.net/forum?id=pc6BgWrCjp*

## Abstract

Deriving meaningful representations from complex, high-dimensional data in unsupervised settings is crucial across diverse machine learning applications. This paper introduces a framework for multi-scale graph network embedding based on spectral graph wavelets that employs a contrastive learning approach. We theoretically show that in Paley-Wiener spaces on combinatorial graphs, the spectral graph wavelets operator provides greater flexibility and control over smoothness compared to the Laplacian operator, motivating our approach. A key advantage of the proposed embedding is its ability to establish a correspondence between the embedding and input feature spaces, enabling the derivation of feature importance. We validate the effectiveness of our graph embedding framework on multiple public datasets across various downstream tasks, including clustering and unsupervised feature importance.

## 1 Introduction

Graph Embeddings and Manifold Learning (Roweis & Saul (2000); Belkin & Niyogi (2003); Tenenbaum et al. (2000); van der Maaten & Hinton (2008); Coifman et al. (2005)) play a pivotal role in analyzing complex data structures encountered in a wide range of machine learning applications. The representations learned by these techniques uncover the intrinsic geometry of the data and support downstream tasks such as clustering and visualization—especially in scenarios where labels are unavailable or unreliable. Most manifold learning methods typically perform nonlinear dimensionality reduction to embed data into lower-dimensional spaces, while some also explore high-dimensional embeddings (Gama et al. (2019); Sun et al. (2009)). Nonetheless, most such methods lack a direct connection to the input features, preventing the evaluation of individual feature contributions and resulting in embeddings with limited interpretability. This limitation is critical in domains like finance, where key behavioral indicators must be traced back to individual features, or in biology, where interpretability is vital for uncovering mechanisms within complex systems. To address these limitations, this paper makes three primary contributions:

- We introduce a framework that leverages multi-scale graph representation using a 3D tensor and contrastive learning. This enhances the expressiveness of the embedding by jointly optimizing over both low- and high-frequency components, enabling the capture of fine-grained data structure.

- We propose a manifold-based interpretation method for feature importance, allowing each dimension of the learned embedding to be directly linked back to a unique original feature.

---

[*]Corresponding author.
[†]Equal second authors.
[§]This work was conducted at Discover Financial Services.

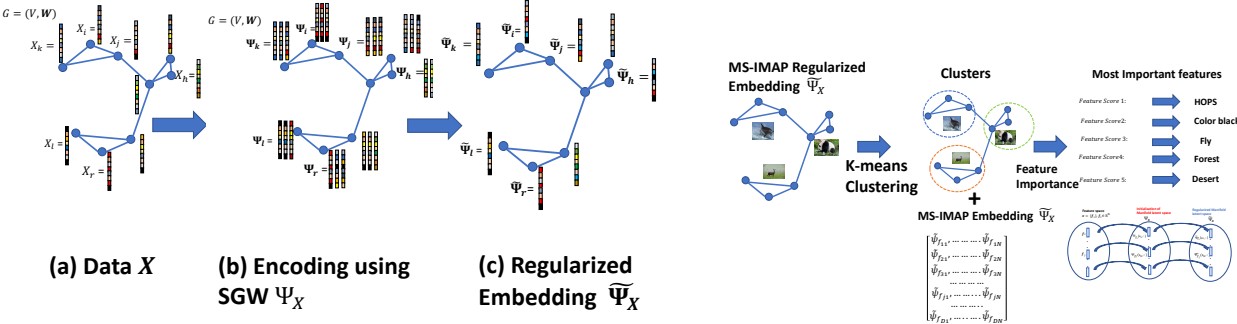

Figure 1: Illustration of the proposed method. (Left) Initial encoding using SGW in (b) and regularized embedding in (c). (Right) Feature embedding space representation, which has a one-to-one correspondence with the input features, can be used along with clustering for interpretability models, such as feature importance.

- We characterize the theoretical representational capacity of the Spectral Graph Wavelet (SGW) (Hammond et al. (2011)) operator by analyzing functions from Paley-Wiener spaces (Pesenson (2008)) on combinatorial graphs. This analysis demonstrates that SGW provides greater flexibility and smoother control compared to traditional Laplacian-based operators.

Our method, MS-IMAP (Multi-Scale Interpretable Manifold Approximation and Projection), is designed to associate each embedding coordinate with a unique input feature, thereby enabling the estimation of feature importance relative to the learned embedding space. This interpretability arises from a deliberate design choice: each embedding coordinate corresponds to a single, unique input feature and is optimized through spectral graph wavelets to align with the underlying graph structure. Unlike standard manifold learning and graph embedding methods, where each embedding dimension typically blends information from all input features, MS-IMAP preserves the semantic identity of each feature by assigning it to a distinct embedding dimension.

Another key distinction of our approach is how it uses the graph Laplacian. Classical methods such as Laplacian Eigenmaps (Belkin & Niyogi (2003)) and many modern techniques such as UMAP (McInnes et al. (2018)) focus mainly on low-frequency components of the Laplacian and therefore emphasize coarse structural information. In contrast, MS-IMAP relies on multi-scale graph representations via SGW within a contrastive learning framework. This design allows the embedding to retain the original dimensionality while still preserving a clear feature-wise interpretation.

Optimization is performed with stochastic gradient descent (SGD) on a 3D tensor–based contrastive objective, so that the learned embeddings capture both local and global graph structure. Empirically, we observe that applying standard feature importance criteria to MS-IMAP embeddings yields more informative feature subsets for clustering than applying the same criteria directly to the raw input features. This indicates a stronger semantic alignment between the embedding and the underlying data structure, which is often lacking in nonlinear deep models or conventional manifold-learning methods. MS-IMAP achieves interpretable and semantically faithful embeddings, while remaining competitive with state-of-the-art graph embedding techniques across a range of tasks.

## 2 Related Work

Extensive efforts have been dedicated to exploring manifold learning methods that aim to produce low-dimensional embeddings while preserving the intrinsic structure of high-dimensional data. Classical manifold learning techniques include Locally Linear Embedding (LLE) (Roweis & Saul (2000)), Laplacian Eigenmaps (Belkin & Niyogi (2003)), Isomap (Tenenbaum et al. (2000)), Diffusion Maps (Coifman et al. (2005)), and

t-distributed Stochastic Neighbor Embedding (t-SNE) (van der Maaten & Hinton (2008)). One of the central challenges in these methods is maintaining a balance between local and global structure, especially in the presence of noise. To address this, various denoising strategies have been developed (Hein & Maier, 2006; Deutsch & Medioni, 2017; Deutsch et al., 2018; Deutsch & Medioni, 2015a;b), aiming to improve robustness and extend the applicability of these techniques to real-world, noisy datasets.

UMAP (McInnes et al. (2018)) has shown strong empirical performance for visualization and cluster separation. It typically employs Laplacian Eigenmaps initialization, which emphasizes low-frequency graph signals while neglecting higher-frequency patterns that may carry important information.

Recent work (Damrich & Hamprecht (2021)) has shown that negative and positive sampling in UMAP significantly influences the effective loss function. These insights have led to studies on alternative sampling schemes (Yan et al. (2023)), bounds on generalization error in embedding methods (Suzuki et al. (2021)), and biases introduced by high-degree nodes during graph construction and sampling (Kojaku et al. (2021)).

Manifold learning and non-linear dimensionality reduction serve purposes beyond visualization; one of their key goals is to learn representations that capture intrinsic geometric structure and approximate geodesic distances. The learned embedding space dimensionality can go beyond two or three dimensions. However, increasing dimensionality does not necessarily improve robustness: noise and the curse of dimensionality can introduce instability. These methods also suffer from computational limitations. Moreover, to compute embedding in higher dimensions many algorithms, such as Laplacian Eigenmaps and UMAP require computing large portions of the Laplacian spectrum, making them computationally intensive for large datasets. Our method addresses these issues by producing interpretable and efficient embeddings while leveraging spectral information across multiple frequency scales.

A key limitation of many manifold learning and graph embedding methods is the absence of an explicit and interpretable mapping, which prevents evaluating the contribution of each input feature to the resulting low-dimensional embedding. This gap restricts their utility in domains where interpretability is crucial. Providing such a link can enable improved feature selection, clearer interpretation of embeddings, and greater robustness to noise (Hosseini & Hammer (2020); Frye et al. (2021); Zharmagambetov & Carreira-Perpinan (2022)).

SGW are central to our method (see Section 3.1), offering localization in both spectral and vertex domains. SGW apply a family of bandpass filters to the graph Laplacian's spectrum, allowing for multi-resolution analysis of graph signals and providing richer representations across frequency bands. Our approach is also related to graph embedding methods that exploit multiscale structure, such as Diffusion Wavelets (Ronald R. Coifman (2006)), Geometric Scattering for Graph Data Analysis (Gao et al. (2018)), and the Graph Scattering Transform (Gama et al. (2019)), which extract graph features for downstream tasks like classification. These methods often embed data in higher-dimensional spaces to improve representation power.

## 2.1 Preliminaries

Consider a set of points $\mathbf{x} = \{\mathbf{x}_n\}, n = 1, ... N, \mathbf{x}_n \in \mathbb{R}^D$, where each $\mathbf{x}_n \in \mathbb{R}^D$, sampled from an unknown smooth manifold $M \subset \mathbb{R}^D$. To model the geometric structure of the data, we construct an undirected, weighted graph $G = (V, \mathbf{W})$ over the point set $\mathbf{x}$, where $V$ is the set of nodes and $\mathbf{W} = (w_{nm})$ is the weighted adjacency matrix encoding edge weights between nodes. The weights $w_{nm}$ can be computed using various techniques. A common approach is to apply a Gaussian kernel to pairs of points $\mathbf{x}_n$ and $\mathbf{x}_m$ within the $k$ nearest neighbors of $\mathbf{x}_n$, denoted as kNN($\mathbf{x}_n$). In this work, we adopt the fuzzy graph in the input space using a smooth exponential kernel (as in McInnes et al. (2018) with a local scaling parameter per point, though other graph construction methods may also be employed. Let $\mathbf{L}$ denote the combinatorial graph Laplacian defined as $\mathbf{L} = \mathbf{D} - \mathbf{W}$, where $\mathbf{D}$ is the diagonal degree matrix with entries $d_{nn} = d(n)$, and the degree $d(n)$ of vertex $n$ is the sum of the weights of all edges incident to $n$. We use the normalized graph Laplacian, defined as: $\mathbf{L}_N = \mathbf{D}^{-1/2}\mathbf{L}\mathbf{D}^{-1/2} = \mathbf{I} - \mathbf{D}^{-1/2}\mathbf{W}\mathbf{D}^{-1/2}$ which has real eigenvalues in the interval $[0, 2]$. For simplicity, we refer to $\mathbf{L}_N$ as $\mathbf{L}$ for the remainder of the paper. Let $\lambda_1, \ldots, \lambda_N$ and $\phi_1, \ldots, \phi_N$ denote the eigenvalues and corresponding eigenvectors of $\mathbf{L}$. We denote the matrix of eigenvectors as $\Phi \in \mathbb{R}^{N \times N}$ and the diagonal matrix of eigenvalues as $\Lambda = \text{diag}(\lambda_1, \ldots, \lambda_N)$. In our setting, each input

coordinate $f_d \in \mathbb{R}^N$ , viewed as a graph signal, is defined across the $N$ nodes of the graph. The graph Fourier transform (GFT) $\hat{f}$ of the graph signal $f$ is defined as the expansion of $f$ in terms of the eigenvectors $\phi$ of the graph Laplacian, so that for frequency $\lambda_l$ we have: $\hat{f}(\lambda_l) = \sum_n f(n)\phi_l^*(n)$.

The original data matrix $\mathbf{x}$ can thus be expressed as a collection of such signals: $\mathbf{x} = (f_1, f_2, .., f_d, ..f_D)$. Our goal is to derive a multi-scale embedding from these signals that approximates the intrinsic manifold coordinates, using spectral information derived from the graph Laplacian.

## 2.2 Multi-scale representations using SGW

In the past two decades, several multi-scale representations for data on irregular graphs have been developed, including Spectral Graph Wavelets (SGW) (Hammond et al. (2011)) and Diffusion Wavelets (Ronald R. Coifman (2006)). In this work, we focus on the SGW-based multi-scale graph transform. SGW offers a principled way to balance spectral and spatial resolution, as its coefficients are localized in both domains. These wavelets are constructed using a kernel function operator $g(\mathbf{L})$, which acts on a signal by modulating each of its spectral components (i.e., Fourier modes). This design enables a trade-off between vertex-domain (spatial) and frequency-domain (spectral) localization. The spatial localization is implicitly controlled by a single scale parameter defined in the spectral domain: greater localization in the vertex domain corresponds to a broader spectral bandwidth. To represent a signal $f \in \mathbb{R}^N$ at multiple scales $S = [s_1, s_2, \ldots, s_K]$, the SGW transform is defined as follows. Let $g(\lambda)$ be a bandpass filter in the spectral domain. Let $\delta_n \in \mathbb{R}^N$ denote the delta function centered at vertex $n \in G$, where $\delta_n(m) = 1$ if $m = n$, and $\delta_n(m) = 0$ otherwise. A wavelet centered at node $n$ and scale $s$ is then given by: $\psi(s, n) = \Phi g(s\lambda)\Phi^T \delta_n$. The value of $\psi(s, n)$ at vertex $m$ can be written explicitly as: $\psi(s, n)(m) = \sum_{l=1}^N g(s\lambda_l)\phi_l^*(n)\phi_l(m)$. Given a graph signal $f \in \mathbb{R}^N$, the **_SGW coefficient_** at node $n$ and scale $s$ is defined as:

$$\psi_f(s, n) = \sum_{l=1}^N g(s\lambda_l)\hat{f}(\lambda_l)\phi_l(n). \tag{1}$$

Each scale $s$ corresponds to a distinct spectral band whose contribution is determined by the SGW filter's response $g(s\lambda_\ell)$ over the Laplacian spectrum, which is adaptive to the graph structure. **Fast computation using Chebyshev polynomials:** Direct computation of SGW coefficients is computationally expensive, requiring $O(N^3)$ operations for $N$ nodes. To address this, SGW proposed an efficient algorithm based on approximating the scaled generating kernels $g(s\lambda)$ using low-order Chebyshev polynomials of the Laplacian $\mathbf{L}$, which are applied directly to the input signal. This significantly reduces computational cost (see Appendix for further details).

## 3 Our Proposed framework: Multi-Scale IMAP

In this section, we introduce Multi-Scale IMAP (MS-IMAP), a framework for interpretable manifold-based embedding that leverages a multi-scale graph representation. The method defines a differentiable mapping $h : \mathbf{x} \mapsto \psi_{\mathbf{x}}$ supported on a discrete graph $G = (V, \mathbf{W})$, where $\psi_{\mathbf{x}}$ denotes a multi-scale graph transform of $\mathbf{x}$. MS-IMAP consists of two main steps.

**Step 1 (multi-scale representation).** We construct a multi-scale graph representation by applying the Spectral Graph Wavelet (SGW) transform to feature signals across multiple scales and graph frequencies. Concretely, we form a 3D tensor of SGW coefficients with dimensions $K \times N \times D$ (number of scales × nodes × features), as detailed in Algorithm 1. This tensor encodes the multi-scale transform in a way that allows us to align the optimized embedding with all scales simultaneously and to enforce a differentiable structure on the transformed features.

The 3D tensor representation is beneficial for two main reasons. First, it enables joint optimization across scales: using the SGW coefficients at all spectral bands within the contrastive loss, we smooth low- and high-frequency information simultaneously over a fixed graph structure. This reduces the need for ad hoc thresholding of high-frequency bands and ensures that all scales are optimized consistently for each pair of nodes. Although neighboring scales may have correlated wavelets, each scale corresponds to a distinct

---

**Algorithm 1:** Encoding multi-scale structure in the optimized embeddings

---

**Input:** Set of points $\{\mathbf{x}_n\}_{n=1}^N$.
**Output:** Initial Node Embeddings $\psi_{\mathbf{x}}$ .
**Step 1:** Construct $G = (V, \mathbf{W})$ from $\{\mathbf{x}_n\}_{n=1}^N$.
**Step 2:** Construct the Laplacian $\mathbf{L}$.
**Step 3:** Compute $\lambda_{\max}(\mathbf{L})$.
**Step 4**: **Initial Embedding construction:**
For $d = 1, .., D$ associated with the feature signals $\{f_d\}$ do:
Compute $\psi_{f_d}(s_k, :)$ at scales $s_k, k = 1, ...K$ using Chebyshev approximation.
**Step 5**: Concatenate $\psi_{f_d}(s_k, :)$ for $d = 1, ..., D$ in a matrix form $\psi_{\mathbf{x}}(s_k, :, :)$ for each fixed scale $s_k$, representing all scales using 3D tensor $\psi_{\mathbf{x}} \in \mathbb{R}^K \times \mathbb{R}^D \times \mathbb{R}^N$.

---

bandwidth and captures graph structure at a different neighborhood radius, so combining them yields a representation that balances local and global structure. Second, the tensor is organized feature-wise, so each embedding coordinate is aligned with a single input feature. This one-to-one correspondence is what underpins the interpretability results in Section 4, where we use the embedding to derive feature-importance rankings.

**Step 2 (embedding optimization).** We then integrate the multi-scale representation into an SGD-based optimization, using a 3D-tensor contrastive objective that leverages both low- and high-frequency information. This optimization refines the multi-scale coefficients into an embedding that is jointly regularized by the graph structure and remains aligned with the original features, leading to fine-grained manifold regularization and improved robustness in downstream tasks.

### 3.1   Step 1: Feature Representation to encode multi-scale structure

To encode multi-scale representations used for subsequent optimization we focus on a 3D tensor based method. Note that each dimension in the embedding space is constructed using a single feature in the original feature set, which is an essential characteristic that can be leveraged for interpreting graph embeddings. **3D-Tensor Encoding**: The 3D-Tensor Encoding method concatenates the multi-scale representation for all features at a fixed scale, generating a matrix representation $\psi_{\mathbf{x}}(s_k, :, :) \in \mathbb{R}^D \times \mathbb{R}^N$ for each scale $s_k$, encoding SGW coefficients across features and nodes. After concatenating all $f_d$ at a fixed scale $s_k$, we have

$$\psi_{\mathbf{x}}(s_k, :, :) = \psi_{f_1}(s_k, :) \| \psi_{f_2}(s_k, :) \| ... \| \psi_{f_D}(s_k, :),$$

where we designate the concatenation using $\|$, with $\mathbf{c}(\psi_{f_i}(s_k, :, :), \psi_{f_j}(s_k, :, :)) = \psi_{f_i}(s_k, :, :) \| \psi_{f_j}(s_k, :, :)$. The optimization can be performed separately for each scale $s_k$, or jointly for all scales $s_k$ using the 3D tensor $\psi_{\mathbf{x}} \in \mathbb{R}^K \times \mathbb{R}^D \times \mathbb{R}^N$ . We detail the proposed feature representation encoding methods in the pseudo code algorithm 1.

### 3.2   Step 2: Optimization design using multi-scale network structure.

We propose to use optimization based on stochastic gradient descent (SGD) that begins with the initial embedding within the spectral graph domain. While the initial spectral graph wavelet (SGW) representation already encodes multi-scale information, it does not yet explicitly incorporate the graph-based structural loss used by MS-IMAP. In this step, we refine the initial embedding by optimizing a multi-scale structural objective across the graph.

**3D-tensor based optimization for graph embedding.** Encoding the representation in a 3D tensor offers two key advantages: (1) it naturally captures the multi-scale structure induced by SGW; and (2) it makes it possible to define a scale-wise structural loss that can be optimized jointly across all nodes and features.

We write the SGW tensor as

$$\psi_{\mathbf{x}}(s_k, d, n) \in \mathbb{R}, \quad k \in \{1, \ldots, K\}, \ d \in \{1, \ldots, D\}, \ n \in \{1, \ldots, N\},$$

where $n, m$ index nodes, $d$ indexes features, and $k$ indexes SGW scales. Equivalently, for each scale $s_k$, we have a slice $\psi_{\mathbf{x}}(s_k, :, :) \in \mathbb{R}^{D \times N}$, whose $d$-th row, $\psi_{\mathbf{x}}(s_k, d, :) \in \mathbb{R}^N$, contains the coefficients of feature $d$ across nodes, and whose $n$-th column, $\psi_{\mathbf{x}}(s_k, :, n) \in \mathbb{R}^D$, contains the coefficients of all features at node $n$.

An SGD update rule at iteration $t$, applied to the 3D tensor, is given by

$$(\tilde{\psi}_{\mathbf{x}}^{(t+1)})_{k,d,n} = (\tilde{\psi}_{\mathbf{x}}^{(t)})_{k,d,n} - \alpha \, \frac{\partial \mathcal{L}}{\partial (\psi_{\mathbf{x}})_{k,d,n}}, \tag{2}$$

where $\alpha$ is the learning rate and $\mathcal{L}$ is the structural loss defined below, and $\tilde{\psi}_{\mathbf{x}}^{(t)} = \psi_{\mathbf{x}}$ for $t = 0$. We employ a cross-entropy loss that compares the empirical graph weights $w_{nm}$ to a similarity term derived from the SGW representation:

$$\mathcal{L}(\tilde{\psi}_{\mathbf{x}} \mid \mathbf{W}) = \sum_{n,m=1}^{N} \sum_{k=1}^{K} \left[ w_{nm} \log \frac{w_{nm}}{v_{nm}^{\psi_{\mathbf{x}}}(s_k)} + \left(1 - w_{nm}\right) \log \frac{1 - w_{nm}}{1 - v_{nm}^{\psi_{\mathbf{x}}}(s_k)} \right], \tag{3}$$

where

$$v_{nm}^{\psi_{\mathbf{x}}}(s_k) = \frac{1}{1 + \left\| \psi_{\mathbf{x}}(s_k, :, n) - \psi_{\mathbf{x}}(s_k, :, m) \right\|_2^2}. \tag{4}$$

Thus, $v_{nm}^{\psi_{\mathbf{x}}}(s_k)$ encodes structural similarity between nodes $n$ and $m$ at scale $s_k$, based on the SGW coefficients of *all* features.

We apply the optimization based on SGD with respect to each scale $s_k$, where the gradient of the loss with respect to the SGW representation at node $n$ can be written in the form

$$\frac{\partial \mathcal{L}\big(\tilde{\psi}_{\mathbf{x}}(s_k, :, :) \mid \mathbf{W}\big)}{\partial \psi_{\mathbf{x}}(s_k, :, n)} = \sum_m w_{nm} \, v_{nm}^{\psi_{\mathbf{x}}}(s_k) \, r_{nm}(s_k) \tag{5}$$

$$- \sum_m \frac{1}{\left\| r_{nm}(s_k) \right\|_2^2} \, v_{nm}^{\psi_{\mathbf{x}}}(s_k) \, r_{nm}(s_k),$$

where

$$r_{nm}(s_k) = \psi_{\mathbf{x}}(s_k, :, n) - \psi_{\mathbf{x}}(s_k, :, m). \tag{6}$$

This expression makes explicit that the gradient at node $n$ depends on SGW differences $r_{nm}(s_k)$ across neighbors $m$ and across all features.

**Feature updates.** Our implementation employs an optimization process that operates jointly across all scales $s_k$ and all nodes, and then aggregates the optimized embeddings with respect to each scale. We define the final embedding tensor as

$$\tilde{\psi}_{\mathbf{x}} = \sum_{k=1}^{K} \tilde{\psi}_{\mathbf{x}}(s_k, :, :), \tag{7}$$

so that the dimensionality of the final embedding is $\tilde{\psi}_{\mathbf{x}} \in \mathbb{R}^{D \times N}$, maintaining a one-to-one correspondence between the $D$ embedding coordinates and the $D$ input features. For each feature index $d$, the $d$-th embedding coordinate is aligned with input feature $d$ and is updated in a *feature-aligned* manner via SGD: the gradient is taken with respect to the entries $\tilde{\psi}_{\mathbf{x}}(s_k, d, n)$, while the structural similarity term $v_{nm}^{\psi_{\mathbf{x}}}(s_k)$ and the graph construction depend jointly on all features through $\psi_{\mathbf{x}}(s_k, :, n)$ and $\psi_{\mathbf{x}}(s_k, :, m)$. Thus, MS-IMAP enforces feature-wise alignment of embedding coordinates with input features, while still incorporating joint structural coupling through the graph and the structural term. This preserves the one-to-one correspondence that is crucial for interpretability.

**Remark 1.** While we adopt the SGW formulation by Hammond et al. (2011), our framework is compatible with any multi-scale spectral graph transform that yields a tensor of coefficients indexed by scales, features,

and nodes. Alternative filter banks or polynomial approximations of the graph Laplacian can be substituted without changing the optimization structure.

**Remark 2.** While the manifold hypothesis suggests that the data lie near a lower-dimensional manifold, this does not require the embedding to be explicitly low-dimensional. In MS-IMAP, we retain the original feature dimensionality to preserve feature-wise interpretability, while still leveraging multi-scale graph structure through the SGW-based optimization. This allows us to avoid repeated large-scale eigenspectrum computations, as in classical methods such as Laplacian Eigenmaps or UMAP, and focus computational effort on the multi-scale structural refinement of the embedding.

## 4  Feature Importance using Explicit Correspondence with Embedding Features

Feature importance is one of the simplest yet most fundamental tools for model interpretability: it associates input features with model outputs or representations, highlighting which features most strongly influence the learned structure. In MS-IMAP, each embedding coordinate has a one-to-one correspondence with an original input feature. This explicit alignment allows us to apply standard feature-importance criteria directly in the embedding space and map the selected dimensions back to specific features.

We show that feature subsets selected from the MS-IMAP embedding lead to superior clustering performance compared to subsets selected from the input space using the same methods. Unlike competing approaches, MS-IMAP retains semantic alignment and supports a direct correspondence between embedding dimensions and original features, making it an interpretable alternative to black-box embeddings. Empirically, applying feature-importance techniques to the MS-IMAP embedding yields more informative feature subsets for clustering than applying the same techniques to the raw input. This provides evidence that our method preserves semantic consistency, a property that traditional nonlinear embeddings typically do not provide. We explore two approaches for estimating feature importance that leverage this explicit correspondence, both in unsupervised or self-supervised settings where feature selection is performed without access to ground-truth labels.

**Laplacian Score (LS) on embedding features.** The first method uses the Laplacian Score (LS) (He et al. (2005)), which assesses the importance of each feature based on how smoothly it varies over a graph constructed from the data. In our setting, the variable corresponding to the $d$-th embedding coordinate, denoted $f_d \in \mathbb{R}^N$, is inferred from the coordinate $\tilde{\psi}_{\mathbf{x}_d}$ in the MS-IMAP embedding. Details on how we compute LS with respect to the embedding features $\tilde{\psi}_{\mathbf{x}}$ are provided in Appendix A.5.

**Mutual-information–based feature importance.** We also consider a self-supervised approach based on Mutual Information (MI). First, we apply $k$-means to the MS-IMAP embedding to obtain clusters, which serve as pseudo-labels. We then compute MI between each embedding feature and these pseudo-labels, analogous to standard MI-based feature selection with true labels. This quantifies how much each feature contributes to cluster assignments and highlights those most informative for distinguishing clusters. The MI-based procedure is summarized in Algorithm 2 (see Appendix A.5).

**Comparison: LS vs. MI-based feature importance.** Laplacian Score and MI-based feature importance are complementary. LS is a fully unsupervised, geometry-driven criterion that favors features which vary smoothly over the graph and align with manifold structure; however, it is not task-aware and can overvalue smoothly varying but irrelevant features. In contrast, MI explicitly measures the statistical dependency between each feature and cluster assignments (obtained from $k$-means on either the raw features or the MS-IMAP embeddings), making it more directly aligned with clustering performance. In our experiments, both criteria become more informative when applied in the MS-IMAP embedding space than in the raw feature space.

## 5  Experimental Results

We evaluated our approach on both synthetic and real datasets in two application domains. First, we demonstrate the use of MS-IMAP embeddings for feature importance, a novel application that highlights

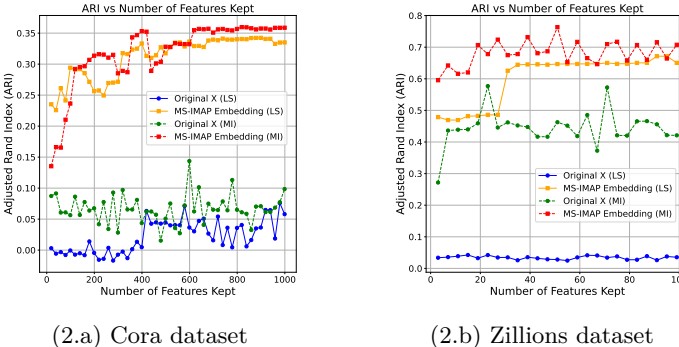

(2.a) Cora dataset          (2.b) Zillions dataset

Figure 2: Clustering results using Adjusted Rand Index (ARI) on Cora and Zillions datasets with a subset of features based on feature importance, comparing the MS-IMAP embedding space and the input features.

the interpretability of our method. Next, we assess the performance of MS-IMAP in clustering tasks using the full embedding space, comparing it against a range of manifold learning-based embedding techniques. **Datasets** We study the performance of MS-IMAP compared to other methods for real datasets. We chose a mix of datasets from varied fields: the Census dataset (Dua & Graff (2017)) is a financial dataset containing information about individuals extracted from the 1994 US Census; the Zilionis dataset (Zilionis et al. (2019)) is a biological dataset containing single-cell sequencing data from different types of cells; the Animals with Attributes (AWA) (Xian et al. (2017)) dataset is an image dataset containing images of animals; The Human Activity Recognition (HAR) (Anguita et al. (2013)) dataset which consists of sensor data collected from subjects performing six different activities:; and the Cora dataset (Sen et al. (2008); McCallum et al. (2000)) is a graph dataset of a citation network consisting of scientific publications.

## 5.1 Assessing Feature Importance via Embedding Correspondence

We demonstrate the application of MS-IMAP for feature importance selection by evaluating clustering performance on subsets of the most significant features. Feature selection is performed with respect to both the input space and the MS-IMAP embedding space, using two alternative methods: Laplacian Score and Mutual Information-based selection. We compare clustering results obtained by applying k-means to subsets of features selected from the embedding, against those selected directly from the original feature space using the same methods. Unlike most manifold learning and graph embedding techniques, MS-IMAP produces embeddings in which each dimension corresponds directly to an input feature. This one-to-one mapping enables direct assessment of individual feature contributions, preserving interpretability.
**Evaluation metrics:** Clustering performance is measured using Adjusted Rand Index (ARI) and Adjusted Mutual Information (AMI). In all experiments, $k$-means is applied in the embedding space. On the Cora and Zillions datasets, Figures 2.a, 2.b, 6.a, and 6.b show that, for both feature selection methods and for both evaluation metrics, clustering accuracy is consistently higher when using the MS-IMAP embedding compared to the original features across varying feature subset sizes. This improvement highlights the robustness of the MS-IMAP embedding, while maintaining interpretability through its feature-wise alignment. Additional experiments on the AWA and HAR datasets, presented in the Appendix, further confirm these gains. In addition to the ARI and AMI evaluations, we further examine the internal clustering structure induced by the selected feature subsets. As detailed in Appendix A.3, we report the silhouette score computed on these subsets to provide a label-free validation of cluster coherence. This complementary analysis confirms that the subsets selected via MS-IMAP yield more stable and well-separated clusters compared to those from the raw feature space.

## 5.2 Clustering Output Embedding Space

We evaluate our method by testing its clustering performance using the full output embedding and comparing it to several representative methods, including UMAP, t-SNE, Diffusion Maps, ISOMAP, PHATE (Moon

et al. (2019)), PaCMAP (Wang et al. (2021)), TriMap (Amid & Warmuth (2019)) and HeatGeo (Huguet et al. (2023)). While most of these methods perform dimensionality reduction, we emphasize that dimensionality reduction is a methodological design choice, not an inherent advantage. Many manifold learning techniques default to low-dimensional embeddings (e.g., 2D or 3D) due to computational constraints such as eigendecomposition, yet this often comes at the cost of reduced expressiveness and, particularly for nonlinear methods, diminished interpretability. However, since most baseline manifold learning methods we compare rely on nonlinear dimensionality reduction, we include clustering results based on the embedding space of the Graph Scattering Transform (Gama et al. (2019)) for completeness. The Graph Scattering Transform, which is primarily used in graph classification and semi-supervised learning tasks, can be adapted for unsupervised learning via simple vector-based concatenation. We note that the Graph Scattering Transform offers less interpretability than MS-IMAP, which preserves a one-to-one correspondence between input features and embedding dimensions. Additionally, the increased dimensionality of the embeddings produced by the Graph Scattering Transform can lead to higher computational costs. As shown in Table 1, MS-IMAP consistently delivers stable and competitive performance, ranking best or second-best across all benchmarks. In contrast, state-of-the-art methods often underperform or show instability on at least one dataset.

**Runtime and Computational Complexity** The computational complexity of Multi-Scale IMAP is of $O(ND \log(N))$. Experiments were performed on Virtual Server containers with 32 cores Intel Xeon 8259CL running at 2.50Ghz and 256GB of RAM. Our method is practical for handling datasets with millions of points and dozens to hundreds of features within a few hours, making the method well-suited for real-world applications. For example, on the Cancer QC data-set of 48,969 samples and 306 features took 27 mins. Additional details and running times on all datasets are provided in the Appendix. Note that the increase in running times compared to methods such as UMAP is primarily due to our extension of manifold learning to include interpretability.

| Data | HAR | | Census | | Zilionis | | AWA | | Cora | |
|---|---|---|---|---|---|---|---|---|---|---|
| Method | ARI | AMI | ARI | AMI | ARI | AMI | ARI | AMI | ARI | AMI |
| Features | 0.45±0.00 | 0.58±0.00 | 0.11±0.00 | 0.15±0.00 | 0.48±0.04 | 0.63±0.01 | 0.07±0.00 | 0.16±0.00 | 0.01±0.02 | 0.07±0.02 |
| ISOMAP | 0.55±0.00 | 0.65±0.00 | 0.18±0.00 | 0.09±0.00 | 0.44±0.03 | 0.56±0.00 | 0.52±0.00 | 0.60±0.00 | 0.12±0.02 | 0.17±0.02 |
| DM | 0.30±0.00 | 0.49±0.00 | 0.01±0.00 | 0.02±0.00 | - | - | 0.22±0.00 | 0.42±0.00 | 0.06±0.00 | 0.07±0.00 |
| UMAP | 0.61±0.00 | 0.73±0.00 | 0.23±0.00 | 0.15±0.00 | 0.52±0.00 | 0.71±0.00 | 0.74 ±0.00 | **0.82**±0.00 | **0.33**±0.01 | **0.36**±0.02 |
| t-SNE | 0.62±0.00 | 0.70±0.00 | 0.20±0.00 | **0.24**±0.00 | 0.39±0.01 | 0.69±0.00 | 0.71±0.00 | 0.80±0.00 | 0.20±0.00 | 0.24±0.00 |
| Scattering.T | 0.45±0.00 | 0.58±0.00 | 0.12±0.00 | 0.16±0.00 | 0.46±0.04 | 0.64±0.02 | 0.67±0.00 | 0.75±0.00 | 0.06±0.03 | 0.14±0.03 |
| PHATE | 0.55±0.02 | 0.66±0.02 | 0.20±0.00 | 0.09±0.01 | 0.65±0.01 | 0.72±0.00 | 0.66±0.03 | 0.73±0.00 | 0.12±0.01 | 0.22±0.01 |
| TriMap | 0.59±0.02 | 0.72±0.01 | 0.20±0.00 | 0.14±0.00 | 0.62±0.08 | **0.77**±0.01 | 0.74±0.04 | 0.81±0.01 | 0.15±0.00 | 0.20±0.00 |
| PaCMAP | 0.62±0.01 | 0.75±0.08 | **0.24**±0.00 | 0.15±0.00 | 0.55±0.03 | 0.76±0.00 | **0.75**±0.04 | 0.81±0.01 | 0.21±0.01 | 0.26±0.00 |
| HeatGeo | 0.60±0.00 | 0.69±0.00 | 0.15±0.00 | 0.10±0.00 | - | - | 0.65±0.00 | 0.74±0.00 | 0.17±0.00 | 0.20±0.00 |
| MS-IMAP | **0.67**±**0.01** | **0.80**±**0.01** | 0.23±0.00 | 0.15±0.00 | **0.70**±**0.01** | 0.76±0.01 | 0.74±0.00 | 0.81±0.00 | **0.33**±**0.00** | **0.36**±**0.00** |

Table 1: Clustering results comparison using ARI and AMI on the Census, Zilionis, AWA, and Cora datasets. Each entry report the mean clustering accuracy and standard deviation. The best performance is **bolded**.

# 6 Theoretical results: Sampling set for Smooth Manifolds with functions defined over Paley-Wiener Spaces

In this section, we characterize the theoretical properties of the representation power of the SGW operator by considering functions sampled from the Paley-Wiener spaces Pesenson (2008) on combinatorial graphs. Pesenson introduced the Paley-Wiener spaces and demonstrated that Paley-Wiener functions of low type are uniquely determined by their values on certain subsets, known as uniqueness sets $U$. We show that the SGW operator can represent functions $f$ within the Paley-Wiener space more efficiently than the graph Laplacian operator $\mathbf{L}$. This efficiency is demonstrated by showing that the SGW operator is more effective in representing functions with larger bandwidth $\omega$ in the Paley-Wiener spaces (i.e., with higher frequencies) using subsets of nodes from the uniqueness sets $U$. To characterize the representation properties of functions defined over $PW_\omega(G)$, we employ the notion of the $\Lambda$-set, introduced by Pesenson which is central to our investigation. Formally, the Paley-Wiener space of $\omega$-bandlimited signals is defined as $PW_\omega(G) = \left\{ f \mid \hat{f}(\lambda) = 0 \ \forall \ \lambda > \omega \right\}$.

The space $L_2(G)$ is defined as the Hilbert space of all complex-valued functions, and $L_2(S)$ is defined as the space of all functions from $L_2(G)$ with support in $S$: $L_2(S) = \{\varphi \in L_2(G) \mid \varphi(v) = 0, v \in V(G) \setminus S\}$. The $\Lambda$-set is defined as follows: a set of vertices $S \subset V$ is a $\Lambda(S)$-set if any $\varphi \in L_2(S)$ satisfies the inequality $||\varphi|| \le \Lambda ||\mathbf{L}\varphi||$ for some constant $\Lambda(S) > 0$. The infimum of all $\Lambda > 0$ for which $S$ is a $\Lambda$-set is denoted by $\Lambda$. Our theoretical results analyze how efficiently the SGW operator can represent band-limited functions on a graph in terms of how many sampled nodes are needed for stable reconstruction. The $\Lambda_\psi$-set associated with the SGW operator $\psi$ quantifies this sampling requirement: a smaller $\Lambda_\psi$ means that all band-limited graph signals can be recovered from fewer samples while preserving stability, and is therefore desirable. An alternative way to view the efficiency of the SGW representation is that because the $\Lambda_\psi$-set associated with the SGW operator is smaller than the corresponding $\Lambda$-set for the Laplacian, one can stably represent band-limited functions with a higher cutoff frequency $\omega$, since the admissible bandwidth is inversely proportional to the $\Lambda$-set value. The ability of the SGW operator $\psi$ to efficiently represent functions $f \in PW_\omega(G)$ can be summarized in the following theorem (the proof is provided in the Appendix)., which highlights the role of the $\Lambda_\psi$-set with respect to the operator $\psi$. We show that any signal $f \in PW_\omega(G)$, where $\lambda_1 \le \omega < \Omega_G$ for some $\Omega_G < \lambda_N$, can be uniquely represented by its samples on the uniqueness set $U$ using the SGW operator. Under certain conditions related to the SGW operator, its associated $\Lambda_\psi$-set is smaller than the $\Lambda$-set associated with the Laplacian operator.

**Theorem 1** *Let $G = (V, \mathbf{W})$ be a connected graph with $n$ vertices and $f \in PW_\omega(G)$ for $\lambda_1 \le \omega < \lambda_{max}$. The SGW operator $\psi$ can be constructed such that the set $S$ is a $\Lambda_\psi$-set and the set $U = V \setminus S$ is a uniqueness set for any space $PW_\omega(G)$ with $\omega < 1/\Lambda_\psi$ and $\Lambda_\psi < \Lambda$ for any $\varphi \in L_2(S)$, where $\Lambda$ is the $\Lambda$-set of the Laplacian operator.*

## 7 Discussion

Identifying the key factors in high-dimensional datasets is essential for leveraging unlabeled data in many practical applications. In this work, we introduce a novel contrastive learning framework for interpretable manifold learning via graph embeddings, leveraging both low- and high-frequency information to enhance representation quality. Our method employs Spectral Graph Wavelets (SGW) to construct a multi-scale graph representation of the input feature space. This representation is optimized using a stochastic gradient descent (SGD)-based scheme, combined with a novel 3D tensor encoding strategy that enriches the embedding process.

MS-IMAP differs fundamentally from traditional dimensionality reduction or visualization techniques. While many manifold-learning methods (e.g., Laplacian Eigenmaps, Diffusion Maps, UMAP) aim to reveal intrinsic structure through low-dimensional embeddings, MS-IMAP instead constructs structure-preserving, interpretable embeddings for downstream tasks by enforcing a one-to-one correspondence between input features and embedding dimensions. The advantage of MS-IMAP does not come from dimensionality alone: simply increasing the number of embedding coordinates often does not lead to an improved performance, as shown both in our comparison with graph scattering transforms and in the additional high-dimensional baselines reported in Appendix A.2.

To enhance the embedding with structural information, we introduce a 3D tensor that captures SGW coefficients across nodes, features, and frequency bands. This enables the model to encode both global and local graph structure. Importantly, although SGW captures information across the graph, each embedding coordinate remains feature-aligned (one per input feature) while its update incorporates joint structural coupling through the shared graph and SGW operator. This preserves the interpretability and separability of features throughout training. We demonstrate the utility of these interpretable embeddings for both feature selection and traditional clustering tasks. Specifically, we assess clustering performance based on features selected from both the input space and the embedding space, using two unsupervised methods: Laplacian Score and Mutual Information-based selection. A key advantage of our framework is its inherent linkage between the original and embedded feature spaces, a property rarely offered by existing nonlinear manifold learning techniques. Furthermore, we show that the resulting embeddings are highly competitive with state-of-the-art graph embedding methods.

To further justify the use of SGW for representation, we analyze its theoretical properties by studying functions in Paley-Wiener spaces on combinatorial graphs. We show that the SGW operator enables more expressive representations, characterized through the concept of the $\Lambda$-set.

**Limitations and future directions include several aspects.** First, our approach assumes that the input features—and the similarity measures used—adequately capture the underlying manifold structure, particularly geodesic distances. Second, extending the current SGD-based optimization framework to handle out-of-sample generalization remains a challenge. Finally, while the embedding-feature correspondence facilitates global feature importance analysis, further improvements may be achieved by incorporating localized importance estimation techniques.

### Acknowledgments

We thank the anonymous reviewers and the Action Editor for their helpful comments and suggestions, which improved the quality of this paper.

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

Table 2: Summary of main notation.

| Symbol | Description |
| --- | --- |
| $N$ | Number of nodes (samples) |
| $D$ | Number of features (input dimensions) |
| $K$ | Number of SGW scales |
| $n, m \in \{1, \ldots, N\}$ | Node (sample) indices |
| $d \in \{1, \ldots, D\}$ | Feature indices |
| $k \in \{1, \ldots, K\}$ | Scale indices |
| $\mathbf{x} \in \mathbb{R}^{N \times D}$ | Input data matrix (nodes $\times$ features) |
| $L$ | Graph Laplacian |
| $L = U \Lambda U^\top$ | Laplacian eigendecomposition |
| $U = [u_0, \ldots, u_{N-1}]$ | Matrix of Laplacian eigenvectors |
| $\Lambda = \mathrm{diag}(\lambda_0, \ldots, \lambda_{N-1})$ | Diagonal matrix of eigenvalues |
| $\psi_{\mathbf{x}}(s_k, d, n)$ | SGW coefficient at scale $s_k$, feature $d$, node $n$ |
| $\psi_{\mathbf{x}}(s_k, :, :) \in \mathbb{R}^{D \times N}$ | SGW tensor slice at scale $s_k$ (all features, all nodes) |
| $\psi_{\mathbf{x}}(s_k, d, :) \in \mathbb{R}^N$ | SGW coefficients of feature $d$ across nodes at scale $s_k$ |
| $\psi_{\mathbf{x}}(s_k, :, n) \in \mathbb{R}^D$ | SGW coefficients of all features at node $n$ at scale $s_k$ |
| $\tilde{\psi}_{\mathbf{x}}(s_k, :, :)$ | Optimized embedding at scale $s_k$ |
| $\tilde{\psi}_{\mathbf{x}} \in \mathbb{R}^{D \times N}$ | Final MS-IMAP embedding (aggregated over scales) |
| $w_{nm}$ | Edge weight between nodes $n$ and $m$ |
| $v_{nm}^{\psi_{\mathbf{x}}}(s_k)$ | SGW-based structural similarity between $n$ and $m$ at scale $s_k$ |

## A   Appendix

### A.1   Theoretical results: Sampling set for Smooth Manifolds with functions defined over Paley-Wiener Spaces

In this section, we characterize the theoretical properties of the representation power of the SGW operator by considering functions sampled from the Paley-Wiener spaces (Pesenson (2008)) on combinatorial graphs. The Paley-Wiener spaces were introduced on combinatorial graphs in (Pesenson (2008)) and a corresponding sampling theory was developed which resembles the classical one. Pesenson proved in (Pesenson (2008)) that Paley-Wiener functions of low type are uniquely determined by their values on certain subgraphs (which are composed from a set of nodes known as the uniqueness sets) and can be reconstructed from such sets in a stable way. We demonstrate that the SGW operator can represent functions $f$ that reside in the Paley-Wiener space on combinatorial graphs more efficiently than the graph Laplacian operator $\mathcal{L}$. The effectiveness of the SGW operator representation in this case can be understood in several ways. In one way, by the ability of the SGW operator to accurately represent functions with larger bandwidth, i.e., $f \in PW_{\omega'}(G)$ where $\omega < \omega'$. We first summarize the main notions and definitions. The space $L_2(G)$ is the Hilbert space of all complex-valued functions $f : V \to \mathbb{C}$ with the following inner product $\langle f, g \rangle = \sum_{v \in G} f(v)\overline{g(v)}$ and the norm

$$||f|| = \left( \sum_{v \in V} |f(v)|^2 \right)^{1/2}. \tag{8}$$

The Laplace (normalized) operator $\mathcal{L}$ is defined by the formula (Pesenson (2008)):

$$\mathcal{L}f(v) = \frac{1}{\sqrt{d(v)}} \sum_{u \sim v} \left( \frac{f(v)}{\sqrt{d(v)}} - \frac{f(u)}{\sqrt{d(u)}} \right), f \in L_2(G) \tag{9}$$

In order to prove these results, we will first need the following definitions:

**Definition 1** *The Paley-Wiener space of $\omega$ -bandlimited signals $f \in L_2(G)$ is defined as follows:*

$$PW_\omega(G) = \left\{ f | \hat{f}(\lambda) = 0 \ \forall \ \lambda > \omega \right\}. \tag{10}$$

We consider a simple, undirected, unweighted, and connected graph $G = (V, \mathbf{W})$, where $V$ is its set of $N$ vertices and $\mathbf{W}$ is its set of edges. The degree of $v$ is number of vertices adjacent to a vertex $v$ is and is denoted by $d(v)$. We assume that degrees of all vertices are bounded by the maximum degree denoted as

$$d(G) = \max_{v \in V} d(v). \tag{11}$$

The following definition (Pesenson (2008)) explains the uniqueness set.

**Definition 2** *A set of vertices $U \subset V$ is a uniqueness set for a space $PW_\omega(G)$ if for every two functions from $PW_\omega(G)$ that coincide on $U$, then they coincide on $V$.*

**Definition 3** *For a subset $S \subset V$, denote $L_2(S)$ as the space of all functions from $L_2(G)$ with support in $S$:*

$$L_2(S) = \left\{ \varphi \in L_2(G), \varphi(v) = 0, v \in V(G) \setminus S \right\}.$$

**Definition 4** *Pesenson (2008) We say that a set of vertices $S \subset V$ is a $\Lambda$ -set if for any $\varphi \in L_2(S)$ it admits a Poincare inequality with a constant $\Lambda = \Lambda(S) > 0$*

$$||\varphi|| \le \Lambda ||\mathcal{L}\varphi||, \ \varphi \in L_2(S). \tag{12}$$

*The infimum of all $\Lambda(S) > 0$ for which $S$ is a $\Lambda$ -set will be called the Poincare constant of the set $S$ and denoted by $\Lambda$.*

The definition above provides a tool to determine when bandlimited signals in Paley-Wiener spaces $PW_\omega(G)$ can be uniquely represented from their samples on a given set. The role of $\Lambda$-sets was explained and proved in the following Theorem by Pesenson (Pesenson (2008)), that shows that if $S \subset V$ then any signal $f \in PW_\omega(G)$ can be uniquely represented by its function values in the complement set $U = V(G) \setminus S$:

**Theorem 2** *Pesenson (2008) If $S \subset V$ is a $\Lambda$- set, then the set $U = V(G) \setminus S$ is a uniqueness set for any space $PW_\omega(G)$ with $\omega < \frac{1}{\Lambda}$.*

**Remark:** Note that non-trivial uniqueness sets can not exist for functions from any Paley-Wiener subspace $PW_\omega(G)$ with any $\lambda_0 \le \omega \le \lambda_N$, but they can exist for some range $\lambda_0 \le \omega < \Omega$, as was shown in (Pesenson (2008)).

We state one of our main results, in which we employ the SGW operator $\psi$ to characterize the uniqueness set using the $\Lambda_\psi$-set, therefore extending the $\Lambda$-set concerning the graph Laplacian operator $\mathcal{L}$.

**Theorem 3** *Let $G = (V, \mathbf{W})$ be a connected graph with $N$ vertices. Assume that there exist a set of vertices $S \subset V$ for which the conditions (1)-(2) in Lemma 1 below hold true. Let $\psi$ be the SGW operator using a polynomial $p(\mathcal{L})$ with the coefficients $\{a_k\}_{k=0}^K$ such that $\psi_f = \sum_{k=0}^K a_k \mathcal{L}^k f$. Then, for any $\varphi \in L_2(S)$, we have that the following inequality holds:*

$$||\varphi|| \le \Lambda_\psi ||\psi_\varphi||, \tag{13}$$

*and thus the set $S$ is a $\Lambda_\psi$-set for the operator $\psi$ with $\Lambda_\psi = \frac{1}{\sqrt{\sum_{k=0}^K \frac{a_k^2}{\Lambda^{2k}}}}$.*

We recall the following results from (Pesenson (2008)). Note that (Pesenson (2008)) established the construction of a $\Lambda$-set by imposing specific assumptions on the sets $S$ and $U$. Our result Theorem 3 holds similar assumptions.

**Lemma 1 (Lemma 3.6 of Pesenson (2008))** *Given a connected graph $G = (V, \mathbf{W})$, a set of vertices $S \subset V$, its complement $U = V \setminus S$, for which the following conditions hold true*

1. *For every $s \in S$ there exists $u \in U$ that is a neighbor of $s$, i.e., $w(u, s) > 0$.*

2. *For every $s \in S$ there exists at least one neighbor node $u \in U$ whose adjacency set intersects $S$ only over $s$.*

*Then there exist a set of vertices $S \subset V$ which is a $\Lambda$-set, with $\Lambda = d(G)$.*

We require one more property of the Laplacian operation, Lemma 3 below. To this end, we use a lemma from (Pesenson (2008)),

**Lemma 2 (Lemma 3.9 of Pesenson (2008))** *If $S$ is a $\Lambda$-set, then for any $\varphi \in L_2(S)$ and all $t \geq 0$, $k = 2^l$, $l = 0, 1, 2, ...$*

$$\frac{1}{\Lambda^k} \|\mathcal{L}^t \varphi\| \leq \|\mathcal{L}^{k+t} \varphi\|, \quad \varphi \in L_2(S)$$

Now we are ready to prove a modification of Lemma 3.9 from (Pesenson (2008)):

**Lemma 3** *Let $S$ be a $\Lambda$-set, $\varphi \in L_2(S)$, and let $m \geq 1$ be an integer. Then*

$$\frac{1}{\Lambda^m} \|\varphi\| \leq \|\mathcal{L}^m \varphi\|.$$

**Proof:** Because every positive integer has binary representation, then we have for some $j_i \in \{0, 1, 2, ...\}, i = 1, 2, ..., N$, that

$$m = \sum_{i=1}^{N} 2^{j_i}.$$

Using Lemma 2, let $k = 2^{j_1}$ and $t = 2^{j_2} + \cdots + 2^{j_N}$, so $m = k + t$. Then we have

$$\|\mathcal{L}^m \varphi\| = \|\mathcal{L}^{k+t} \varphi\|$$
$$\geq \frac{1}{\Lambda^k} \|\mathcal{L}^t \varphi\|$$
$$= \frac{1}{\Lambda^{2^{j_1}}} \|\mathcal{L}^{2^{j_2} + \cdots + 2^{j_N}} \varphi\|$$

We can now let $k_2 = 2^{j_2}$ and $t_2 = 2^{j_3} + \cdots + 2^{j_N}$ so that,

$$\|\mathcal{L}^m \varphi\| \geq \frac{1}{\Lambda^{2^{j_1}}} \|\mathcal{L}^{2^{j_2} + \cdots + 2^{j_N}} \varphi\| = \frac{1}{\Lambda^{2^{j_1}}} \|\mathcal{L}^{k_2 + t_2} \varphi\|$$
$$\geq \frac{1}{\Lambda^{2^{j_1}}} \frac{1}{\Lambda^{k_2}} \|\mathcal{L}^{t_2} \varphi\|$$
$$= \frac{1}{\Lambda^{2^{j_1} + 2^{j_2}}} \|\mathcal{L}^{2^{j_3} + \cdots + 2^{j_N}} \varphi\|.$$

Continuing in this way, we eventually get,

$$\|\mathcal{L}^m \varphi\| \geq \frac{1}{\Lambda^m} \|\varphi\|.$$

completing the proof. $\square$

In the next theorem, we expand the characterization of the uniqueness set using the $\Lambda$-set concerning the graph Laplacian operator $\mathcal{L}$ to include cases where we employ the SGW operator $\psi$. We thereby characterize the uniqueness set using the $\Lambda_\psi$-set for the SGW operator.
We now turn to prove Theorem 3, which was stated earlier.
**Proof of Theorem 3:**

Let $S \subset V$ be a set satisfying the conditions in Lemma 1, so $S$ is $\Lambda$-set. Then by Lemma 3, for any integer $k \geq 1$,

$$\frac{1}{\Lambda^k}\|\varphi\| \leq \|\mathcal{L}^k \varphi\|.$$

The SGW operator of $\varphi$ has the form,

$$\psi_\varphi := \sum_{i,j} \sum_{k=0}^{K} a_k \mathcal{L}^k \varphi,$$

for some $K \geq 1$, $a_k \in \mathbb{R}$ are coefficients, and where the summation over $i$ and $j$ represents the summation over the nodes of the graph. Now we calculate,

$$
\begin{aligned}
\|\psi_\varphi\| &= \left( \sum_{i,j} \sum_{k=0}^{K} \left| a_k (\mathcal{L}^k \varphi)_{ij} \right|^2 \right)^{1/2} \\
&= \left( \sum_{i,j} \sum_{k=0}^{K} a_k^2 \left| (\mathcal{L}^k \varphi)_{ij} \right|^2 \right)^{1/2} \\
&= \left( \sum_{k=0}^{K} a_k^2 \sum_{i,j} \left| (\mathcal{L}^k \varphi)_{ij} \right|^2 \right)^{1/2} \\
&= \left( \sum_{k=0}^{K} a_k^2 \|\mathcal{L}^k \varphi\|^2 \right)^{1/2} \\
&\geq \left( \sum_{k=0}^{K} a_k^2 \frac{1}{\Lambda^{2k}} \|\varphi\|^2 \right)^{1/2} \\
&= \|\varphi\| \left( \sum_{k=0}^{K} a_k^2 \frac{1}{\Lambda^{2k}} \right)^{1/2}.
\end{aligned}
$$

Letting

$$\Lambda_\psi := \frac{1}{\sqrt{\sum_{k=0}^{K} \frac{a_k^2}{\Lambda^{2k}}}}.$$

Then we have $\|\varphi\| \leq \Lambda_\psi \|\psi_\varphi\|$, so $S$ is also a $\Lambda_\psi$-set.

$\square$

From the assumptions and proof of Theorem 3, we get the immediate result:

**Corollary 1** *Let $S \subset V$ be a $\Lambda$-set. Then $S$ is also a $\Lambda_\psi$-set.*

**Remark 1:** An important property which can be observed from Theorem 3 is the following: given the SGW $\psi_f$ as a spectral representation operator applied on $f$, we can choose coefficients $\{a_k\}_{k=0}^{K}$ such that we obtain a $\Lambda_\psi$ - set associated with the operator $\psi$, which is smaller than the $\Lambda$ - set associated with the Laplacian operator $\mathcal{L}$. This implies that the operator $\psi$ provides more flexibility and better control over smoothness properties in comparison to the Laplacian operator.

**Remark 2:** Since the $\Lambda_\psi$ - set can be chosen to be smaller than $\Lambda$ - set (for a proper choice of the coefficients $\{a_k\}_{k=0}^{K}$ using the operator $\psi$) then the SGW operator $\psi$ provides a more efficient representation for $f \in PW_{\omega'}(G)$ with $\omega < \omega'$ using the same subsets of nodes from the uniqueness set $U$ in comparison to the Laplacian operator $\mathcal{L}$.

**Remark 3:** Note that the characterization of the the uniqueness set does not rely on a reconstruction method of the graph signal values of $f(S)$ from their known values on $U$.

The next Theorem demonstrates the role of the $\Lambda_\psi$ -set with respect to the operator $\psi$, where we show that any signal $f \in PW_\omega(G)$ can be uniquely represented by its samples on the uniqueness set $U$. This results resembles the role of $\Lambda$-sets with respect to the graph Laplacian operator $\mathcal{L}$, yet with a different bound then Lemma 2.

**Theorem 4** *Let $G = (V, \mathbf{W})$ be a connected graph with $N$ vertices and $f \in PW_\omega(G)$ for $\lambda_1 < \omega < \lambda_{max}$. Given the SGW operator $\psi$, and a set $S$ which is a $\Lambda_\psi$ - set. Then the set $U = V \setminus S$ is a uniqueness set for any space $PW_\omega(G)$ with $\omega < 1/\Lambda_\psi$.*

**Proof:** Given $f, \tilde{f} \in PW_\omega(G)$, then $f - \tilde{f} \in PW_\omega(G)$. Assume that $f \neq \tilde{f}$. If $f, \tilde{f}$ coincide on $U = V \setminus S$, then $f - \tilde{f} \in L_2(S)$ and therefore

$$||f - \tilde{f}|| \leq \Lambda_\psi ||\psi_{f-\tilde{f}}||. \tag{14}$$

Since $\psi_f \in \mathbb{R}^N$, we have that by properties of a vector space in $\mathbb{R}^N$, using the Cauchy–Schwarz inequality and assuming $|a_k| \leq 1 \, \forall k$, we have:

$$||\psi_{f-\tilde{f}}|| \leq \omega ||f - \tilde{f}||. \tag{15}$$

Combining the inequalities above and using the inequality $\Lambda_\psi \omega < 1$ we have that

$$||f - \tilde{f}|| \leq \Lambda_\psi ||\psi_{f-\tilde{f}}|| \leq \Lambda_\psi \omega ||f - \tilde{f}|| < ||f - \tilde{f}|| \tag{16}$$

which is a contradiction to the assumption that $f \neq \tilde{f}$. Thus, the set $U = V \setminus S$ is a uniqueness set for any space $PW_\omega(G)$ with $\omega < 1/\Lambda_\psi$. $\square$

**Remark 1:** Note that $\Lambda \omega < 1$ implies that $\Lambda_\psi \omega < 1$ given $\Lambda_\psi < \Lambda$, then we can increase the size number of nodes in the uniqueness set $U$ for $PW_\omega(G)$ (for a proper choice of the coefficients $\{a_k\}_{k=0}^K$ using the operator $\psi$). In other words, we may increase the size of $S$ (thus reducing the size of $U$) and still obtain a uniqueness set with a smaller size for the graph signals in $PW_\omega(G)$.

**Remark 2:** We note that the results of Theorem 4 concerning the uniqueness set are independent from the stability properties of the representation. In order to achieve stability which is important for reconstruction, it is required to construct a wavelet operator using multiple scales $t_j, j = 1, ...J$, as proposed in (Hammond et al. (2011)). More specifically, we can express the function $\varphi$ using multiple scales $t_j$ (here we replace the previous notation of scale $s_j$ with $t_j, j = 1, ..J$, not to confuse with nodes $s \in S$). Then for a fixed scale $t_j$ we have that the SGW is given by $||\psi_\varphi(t_j)|| = \left( \sum_{i=1}^N \sum_{k=0}^K |a_{t_j,k}(\mathcal{L}^k \varphi(i))|^2 \right)^{1/2}$ and

$$||\psi_\varphi|| = \left( \sum_j^N \sum_{i=1}^N \sum_{k=0}^K |a_{t_j,k}(\mathcal{L}^k \varphi(i))|^2 \right)^{1/2}. \tag{17}$$

In a similar way to the arguments provided in Theorem 3 we can choose coefficients $a_{t_j,k}$ associated with the Laplacian polynomial such that the inequality $||\varphi|| \leq \Lambda_\psi ||\psi_\varphi||$ is satisfied.

**Remark 3:** Optimal Choice of coefficients $a_k$ for Minimizing $\Lambda_\psi$ for optimal choice of $\Lambda_\psi$ in the multi-scale setting.

To find the coefficients $a_{t_j,k}$ that minimize $\Lambda_\psi$ where:

$$\Lambda_\psi = \frac{1}{\sqrt{\sum_{j=1}^J \sum_{k=0}^K \frac{a_{t_j,k}^2}{\Lambda^{2k}}}}, \tag{18}$$

subject to the constraint: $\sum_{j=1}^J \sum_{k=0}^K a_{t_j,k}^2 = 1$, we optimize the coefficients $a_{t_j,k}$, which control how much of the signal at scale $t_j$ and power $k$ is captured by the SGW operator. We use the Lagrangian $\mathcal{F}$,

incorporating the energy constraint into the optimization problem and Define:

$$\mathcal{F}(a_{t_1,0}, \ldots, a_{t_J,K}, \mu) = \frac{1}{\sqrt{\sum_{j=1}^{J} \sum_{k=0}^{K} \frac{a_{t_j,k}^2}{\Lambda^{2k}}}} + \mu \left( \sum_{j=1}^{J} \sum_{k=0}^{K} a_{t_j,k}^2 - 1 \right),$$

(19)

where $\mu$ is the Lagrange multiplier that ensures the total energy of the coefficients is equal to 1.

Taking partial derivatives, solving for $a_{t_j,k}$, and differentiating $\mathcal{L}$ with respect to each coefficient $a_{t_j,k}$:

$$\frac{\partial \mathcal{F}}{\partial a_{t_j,k}} = \frac{-a_{t_j,k}/\Lambda^{2k}}{\left( \sum_{j=1}^{J} \sum_{k=0}^{K} \frac{a_{t_j,k}^2}{\Lambda^{2k}} \right)^{3/2}} + 2\mu a_{t_j,k} = 0.$$

(20)

rearranging the equation above we obtain:

$$\frac{a_{t_j,k}}{\Lambda^{2k}} = 2\mu a_{t_j,k} \left( \sum_{j=1}^{J} \sum_{k=0}^{K} \frac{a_{t_j,k}^2}{\Lambda^{2k}} \right)^{3/2},$$

assuming $a_{t_j,k} \neq 0$, we can divide both sides of the equation by $a_{t_j,k}$, yielding:

$$\frac{1}{\Lambda^{2k}} = 2\mu \left( \sum_{j=1}^{J} \sum_{k=0}^{K} \frac{a_{t_j,k}^2}{\Lambda^{2k}} \right)^{3/2},$$

(21)

thus the sum $\sum_{j=1}^{J} \sum_{k=0}^{K} \frac{a_{t_j,k}^2}{\Lambda^{2k}}$ is proportional to the inverse of $\Lambda^{2k}$, implying that the optimal wavelet coefficients should decay proportionally to the inverse powers of the Laplacian eigenvalues $a_{t_j,k} \propto \frac{1}{\Lambda^k}$.

Next, applying the energy constraint: $\sum_{j=1}^{J} \sum_{k=0}^{K} a_{t_j,k}^2 = 1$ and substituting $a_{t_j,k} = \frac{1}{\Lambda^k}$, we obtain:

$$\sum_{j=1}^{J} \sum_{k=0}^{K} \frac{1}{\Lambda^{2k}} = 1.$$

(22)

Thus, the normalized optimal coefficients $a_{t_j,k}$ are:

$$a_{t_j,k} = \frac{\frac{1}{\Lambda^k}}{\sqrt{\sum_{j=1}^{J} \sum_{k=0}^{K} \frac{1}{\Lambda^{2k}}}}.$$

(23)

Note that the wavelet energy is distributed across all scales and powers of the Laplacian, with a decay that prioritizes low-frequency components (small $k$).

Using the form of the coefficients $a_{t_j,k}$ above, the expression for $\Lambda_\psi$ is:

$$\Lambda_\psi = \frac{1}{\sqrt{\sum_{j=1}^{J} \sum_{k=0}^{K} \frac{1}{\Lambda^{4k}}}},$$

(24)

which is the minimized value of $\Lambda_\psi$, showing that the optimal choice of coefficients reduces the overall constant, improving the localization and efficiency of the SGW operator. Thus, the optimal choice of wavelet coefficients $a_{t_j,k}$ that minimizes $\Lambda_\psi$ in the multi-scale SGW operator is proportional to $\frac{1}{\Lambda^k}$, prioritizing low-frequency components.

### A.2 Additional analysis of embedding dimensionality

This section provides additional clarification and experiments regarding the role of embedding dimensionality and the comparison between MS-IMAP and high-capacity baselines.

**Dimensionality comparison.** As discussed in Sec. 5.2, MS-IMAP is not a dimensionality reduction or visualization method. It is an interpretable graph embedding that preserves a one-to-one correspondence between input features and embedding coordinates by design, and therefore retains the original feature dimensionality rather than compressing to 2–3 dimensions. The comparison in Sec. 5.2 is intended to situate MS-IMAP with respect to standard manifold learning and visualization baselines and to show that, even under the additional constraint of preserving a one-to-one feature–embedding correspondence, its performance remains comparable to these methods. The goal is not to attribute performance solely to higher dimensionality; in fact, increasing dimensionality alone often does not guarantee improved performance, as illustrated both in our comparison with graph scattering transforms and in the additional results reported here.

To more directly assess the effect of representational capacity, we include comparisons to higher-capacity unsupervised representation methods (e.g., autoencoder / self-supervised baselines) and to higher-dimensional variants of manifold learning methods using multiple choices of embedding dimensionality. This allows us to test whether MS-IMAP's benefits persist when competing methods are not constrained to 2–3 dimensions and to make the role of dimensionality more transparent.

We specifically examine:

1. **Unsupervised deep representations (GAE).** We add a Graph Auto-Encoder (GAE) baseline following (Kipf & Welling (2016)), implemented via PyTorch Geometric's `GAE` class, with a 2-layer GCN encoder (Kipf and Welling, 2017) and an inner-product decoder trained in a self-supervised fashion to reconstruct the $k$NN adjacency matrix built from the input features. We tune the embedding dimensionality over

$$d \in \{2, 5, 10, 20, 50, D\},$$

   where $D$ is the input dimensionality, and report in Table 3 the best clustering performance (ARI/AMI) achieved by GAE over these values of $d$ for each dataset.

2. **Higher-dimensional manifold learning baselines.** We likewise re-run the manifold learning methods UMAP, HeatGeo, and PHATE with the same set of output dimensions,

$$d \in \{2, 5, 10, 20, 50, D\},$$

   and again report in Table 3 the best performance over $d$ for each method and dataset.

The experimental results show that, in most cases, increasing the embedding dimension beyond the standard 2–3 dimensions does not yield substantial performance gains for these methods. Across datasets, MS-IMAP remains competitive with state-of-the-art baselines, typically achieving the first- or second-best ARI/AMI. For GAE in particular, even with careful tuning of $d$, its clustering performance on these tabular $k$NN-graph datasets is comparable to or below that of simpler manifold-learning baselines. This is consistent with prior observations that unsupervised deep graph encoders can struggle to produce effective embeddings for clustering when the underlying graphs are irregular or constructed from tabular features rather than arising from a natural network structure.

**Remark:** Please note that the discrepancy between Table 1 and Table 3 comes from differences in the evaluation protocol rather than from MS-IMAP itself. In Table 1, each baseline (including UMAP and PHATE) was run with its own hyperparameter sweep tuned for the standard low-dimensional use case (e.g., recommended configurations at 2–3 dimensions), and we reported the best result from that sweep. In Table 3, the goal is specifically to study the effect of embedding dimensionality. To keep this experiment tractable, we fixed the remaining hyperparameters across the different values of ($d \in 2, 5, 10, 20, 50, D$) (and, for some methods, averaged over multiple random seeds) rather than performing a full hyperparameter search for

Table 3: Clustering performance (ARI / AMI) for the dimensionality experiment. For each method and dataset we report the best score over $d \in \{2, 5, 10, 20, 50, D\}$.

| Method | HAR | | Census | | Zillionis | | AWA | | Cora | |
|---|---|---|---|---|---|---|---|---|---|---|
| | ARI | AMI | ARI | AMI | ARI | AMI | ARI | AMI | ARI | AMI |
| UMAP | 0.64 | 0.76 | 0.22 | 0.14 | 0.61 | **0.76** | **0.8** | **0.82** | **0.35** | **0.38** |
| HeatGeo | 0.63 | 0.71 | N/A | N/A | N/A | N/A | 0.77 | **0.82** | 0.18 | 0.22 |
| PHATE | 0.56 | 0.67 | 0.2 | 0.092 | 0.27 | 0.71 | 0.76 | 0.79 | 0.16 | N/A |
| GAE | 0.39 | 0.32 | 0.05 | 0.09 | 0.53 | 0.64 | 0.7 | 0.77 | 0.252 | 0.30 |
| MS-IMAP | **0.67** | **0.80** | **0.23** | **0.15** | **0.70** | **0.76** | 0.74 | 0.81 | 0.33 | 0.36 |

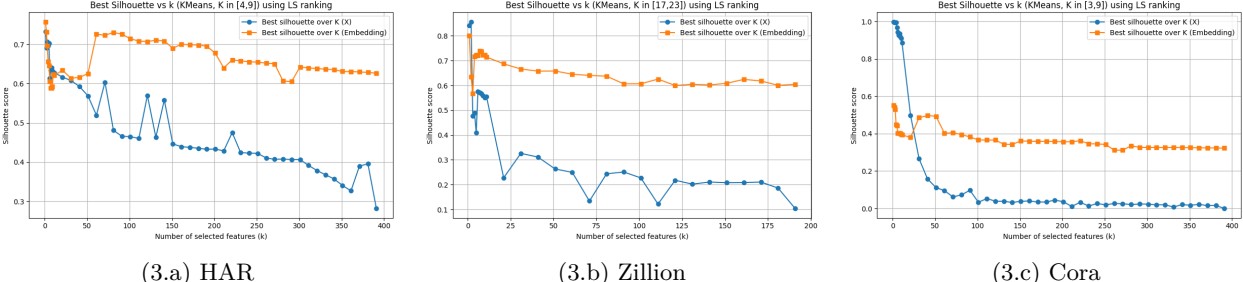

| (3.a) HAR | (3.b) Zillion | (3.c) Cora |
|---|---|---|

Figure 3: Silhouette score as a function of the number of selected features $k$ on HAR, Zillion, and Cora, comparing the label-free clustering quality obtained from the raw features (blue) and from the MS-IMAP embedding (orange), both ranked by Laplacian Score. For each dataset and each $k$, we run $k$-means with several choices of the number of clusters $C$ and report the best silhouette score over $C$.

every method, $d$ and dataset combination. As a consequence, the exact configuration that achieved the best scores in Table 1 is not guaranteed to appear among the Table 3 runs, so the "best over ($d$)" values in Table 3 can be slightly lower than the Table 1 numbers for some methods and datasets. Thus Table 3 show that under a consistent and reasonably tuned setup across dimensions, increasing the embedding dimension beyond 2–3 does not systematically improve the performance of these baselines, and that MS-IMAP remains competitive in this higher-capacity regime.

Finally, the choice to retain the input dimensionality in MS-IMAP is primarily motivated by interpretability: most nonlinear embeddings sacrifice the direct, feature-wise correspondence that is central to the feature-importance analyses in Secs. 4–5.1. The results in Table 3 indicate that this interpretability does not come at the cost of substantial performance loss.

### A.3 Silhouette-based analysis of feature selection

To further analyze the effect of feature selection on clustering structure, we report the internal clustering metric of the silhouette score on the selected feature subsets. This label-free measure does not use ground-truth labels and therefore does not replace our primary ARI/AMI evaluation, but it provides an additional unsupervised check that the selected subsets induce coherent clusters in the embedding space. For each dataset and each number of selected features $k$, we run $k$-means on the selected features for several values of the cluster number $C$ (e.g., $C \in [4, 9]$ for HAR, $C \in [7, 13]$ for AWA, $C \in [2, 4]$ for Census) and record the best silhouette score over $C$.

Figures 3–4 show the resulting curves when Laplacian Score is applied either to the original features or to the MS-IMAP embedding. For HAR, Zillion, AWA, and Census, the MS-IMAP embedding consistently achieves higher and more stable silhouette scores (typically around 0.6–0.7) than the original features, whose scores degrade steadily as $k$ increases. For Cora, the original feature space exhibits a very high silhouette for the top one or two features but then rapidly collapses toward values near zero as more features are added,

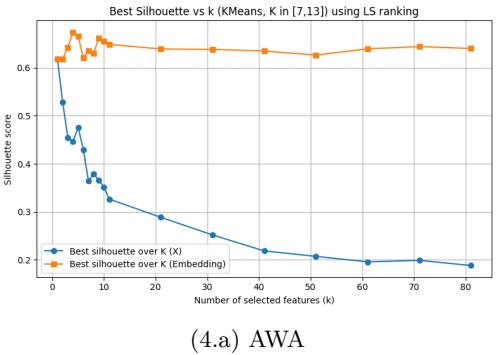
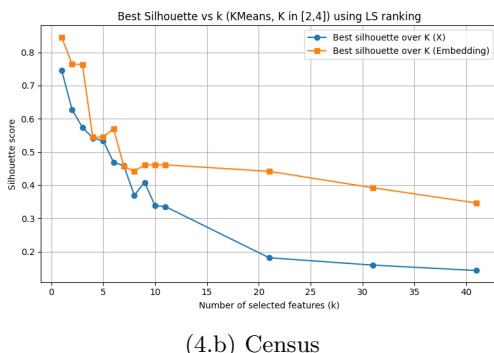

(4.a) AWA

(4.b) Census

Figure 4: Silhouette score as a function of the number of selected features $k$ on AWA and Census, comparing the label-free clustering quality obtained from the raw features (blue) and from the MS-IMAP embedding (orange), both ranked by Laplacian Score. As in Fig. 3, for each dataset and each $k$ we report the best silhouette score over several choices of the number of clusters $C$.

whereas the MS-IMAP embedding maintains a moderate but stable silhouette (around 0.3) over a wide range of $k$. Overall, MS-IMAP–based feature selection trades a narrow silhouette peak for a small number of raw features for more robust and consistently higher cluster quality once $k$ exceeds a few features. A systematic study of model-selection strategies for choosing $C$ or the embedding dimension (e.g., eigengap or silhouette-based selection) in combination with MS-IMAP is left for future work.

### A.4 Alternative Encoding: Encoding By Simple concatenation

An alternative to the 3D tensor-based representation approach presented in the main body of the paper, one can use simple concatenation, where all features are transformed using SGW and each data point is represented by a vector formed by concatenating all corresponding SGW coefficients. For completeness, we also provide a simple encoding method that involves concatenating the multi-scale representation of all features and filters (associated with different scales) into a single vector representation for each point. This results in a matrix representation denoted as $\psi_{\mathbf{x}}$. We detail the encoding and optimization employed for this method. Note that we designate the concatenation using $||$, with

$$\mathbf{c}(\psi_{f_i}(s_k,:), \psi_{f_j}(s_k,:)) = \psi_{f_i}(s_k,:) \,||\, \psi_{f_j}(s_k,:),$$

denoting the concatenation of the vectors corresponding to the multi-scale representation $\psi_{f_i}(s_k,:)$ and $\psi_{f_j}(s_k,:)$. For method 1 all features and all scales are concatenated together, the resulting matrix $\psi_{\mathbf{x}}$ can be represented as

$$\psi_{\mathbf{x}} = \psi_{f_1}(s_1,:)||\psi_{f_2}(s_1,:)||....||\psi_{f_{D-1}}(s_K,:)||\psi_{f_D}(s_K,:),$$

where $\psi_{\mathbf{x}} \in \mathbb{R}^{N \times (KD)}$.

1. **Optimize Embedding Method 1:**
Given the encoded multi-scale representation $\psi_{\mathbf{x}} \in \mathbb{R}^{N \times (KD)}$

$\psi_{\mathbf{x}} \in \mathbb{R}^{KD} \times \mathbb{R}^N$ perform optimization in the SGW domain, using the following fuzzy cross entropy loss function:

$$\mathcal{L}(\tilde{\psi}_{\mathbf{x}}|\mathbf{W}) = \sum_{n,m} \left( w_{nm}\log\frac{w_{nm}}{v_{nm}^{\psi_{\mathbf{x}}}} + (1 - w_{nm})\log\frac{1 - w_{nm}}{1 - v_{nm}^{\psi_{\mathbf{x}}}} \right), \tag{25}$$

where $v_{nm}^{\psi_{\mathbf{x}}} = \frac{1}{1+||\psi_{\mathbf{x}_n} - \psi_{\mathbf{x}_m}||^2}$.

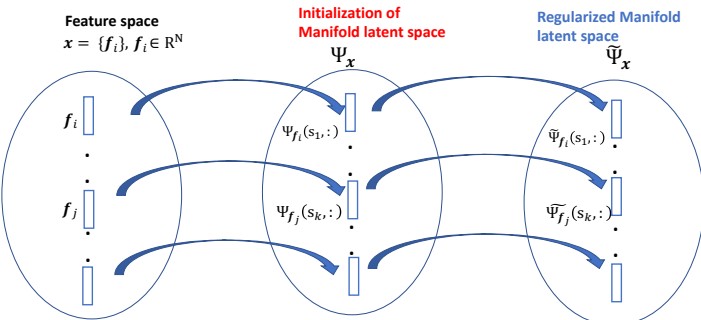

Figure 5: The encoding step of our framework is illustrated as a mapping between each coordinate of the input features and a corresponding dimension in the embedding space. This mapping is facilitated through our proposed approach, which aligns the input dimensions of the features with the latent embedding space. By establishing this correspondence, we acquire the capability to interpret each dimension in the embedding space. Such interpretation allows for various analyses, including providing feature importance relative to the latent space.

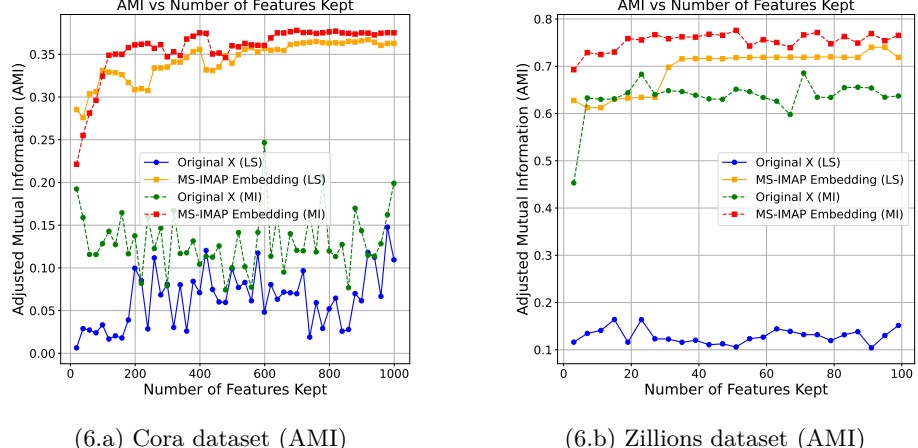

(6.a) Cora dataset (AMI)  (6.b) Zillions dataset (AMI)

Figure 6: Clustering results on Cora and Zillions datasets using a subset of features based on feature importance, comparing the MS-IMAP embedding space and the input features. Plots (a)–(b) show Adjusted Mutual Information (AMI) results.

Dropping terms that do not depends on $\psi_{\mathbf{x}_n}$, the gradient of the loss is approximated by:

$$\frac{\partial \mathcal{L}(\tilde{\psi}_{\mathbf{x}}|\mathbf{W})}{\partial \psi_{\mathbf{x}_n}} \sim \sum_m w_{nm} v_{nm}^{\psi_{\mathbf{x}}}(r_{nm}) - \sum_m \frac{1}{||r_{nm}||^2} v_{nm}^{\psi_{\mathbf{x}}}(\psi_{\mathbf{x}_n} - \psi_{\mathbf{x}_m}), \tag{26}$$

where $r_{nm} = \psi_{\mathbf{x}_n} - \psi_{\mathbf{x}_m}$, and $\psi_{\mathbf{x}_n}$ corresponds to the vector associated with node $n$ multi-scale representation concatenated for all the graph signals and all spectral bands.

---

**Algorithm 2:** MI-based Unsupervised Feature Selection

---

**Input:** MS-IMAP embeddings $\tilde{\psi}_\mathbf{x}$, number of clusters $k$
**Output:** MI scores for each feature
**Step 1:** Apply k-means clustering on the embedding space $\psi_\mathbf{x}$ to obtain cluster labels
      clusters = k-means($\tilde{\psi}_\mathbf{x}, k$)
**Step 2:** Use the resulting cluster labels as pseudo-labels
      pseudo_labels = $c(\tilde{\psi}_\mathbf{x})$
**Step 3:** Initialize an empty list to store mutual information (MI) values for each feature
      mi_scores = []
**Step 4:** For each feature $d$ in the original data or embedding space:
      **Step 4.1:** Compute mutual information between feature $\tilde{\psi}_\mathbf{x}(d)$ and the pseudo-labels
      **Step 4.2:** $MI\_score = \mathrm{MI}(\tilde{\psi}_\mathbf{x}(d), c(\tilde{\psi}_\mathbf{x}))$
      **Step 4.3:** Append $MI\_score$ to mi_scores
**Step 5:** Output the list of MI scores
      **Return** mi_scores

---

## A.5 Additional details: Embedding Feature Importance using Mutual Information and Laplacian Score

### A.5.1 Computing the Laplacian Score with respect to the embedding features

The Laplacian score is calculated with respect to embedding features $\tilde{\psi}_\mathbf{x}$ using the Laplacian graph $\mathbf{L}$ and the degree matrix $\mathbf{D}$. To compute the Laplacian Score, we first subtract the mean and then calculate it as follows: $L_s(\tilde{\psi}_\mathbf{x})_d = \frac{(\tilde{\psi}_\mathbf{x})_d^T \mathbf{L}(\tilde{\psi}_\mathbf{x})_d}{(\tilde{\psi}_\mathbf{x})_d^T \mathbf{D}(\tilde{\psi}_\mathbf{x})_d}$. Smaller scores indicate that the embedding feature $(\tilde{\psi}_\mathbf{x})_d$ has a greater projection onto the subspace of eigenvectors corresponding to the smallest eigenvalues, signifying higher importance concerning the global graph structure. Therefore, the feature importance of $(\tilde{\psi}_\mathbf{x})_d$ can be directly interpreted as the importance of the corresponding original feature. Although the graph is typically constructed using local neighborhoods (e.g., via k-nearest neighbors), the Laplacian operator reflects global smoothness over the entire data manifold.

### A.5.2 Embedding Feature Importance using Mutual Information

We summarize the MI-based unsupervised feature selection procedure in Algorithm 2., as described in Section 4. MI based unsupervised feature selection is self-supervised (using the pseudo-labels), capturing which features are useful for distinguishing between clusters. MI scores are also directly interpretable, where a high MI score indicates that the feature contributes to separating groups.

In comparison, applying the Laplacian Score(He et al. (2005)) to the embedding may be less effective for feature importance, as the embedding process can mix original features in ways that obscure their individual contributions. While the Laplacian Score prioritizes local neighborhood preservation, this does not necessarily correspond to discriminative power for clustering. In contrast, the proposed MI-based approach directly identifies features that are informative for distinguishing clusters. However, it relies on the quality of the pseudo-labels derived from clustering, which may not always reflect true cluster structure.

We also include additional experiments that demonstrate the effectiveness of MS-IMAP for feature importance by evaluating clustering performance as a function of the number of selected features—complementing the results presented in Section 4. Specifically, we assess the most significant features identified using Laplacian Score and Mutual Information-based selection, applied to both the MS-IMAP embedding and the original feature space.

Clustering results, evaluated using Adjusted Rand Index (ARI) and Adjusted Mutual Information (AMI), are obtained by applying $k$-means to different subsets of embedding features selected by each method, and compared to the corresponding subsets selected directly from the raw features.

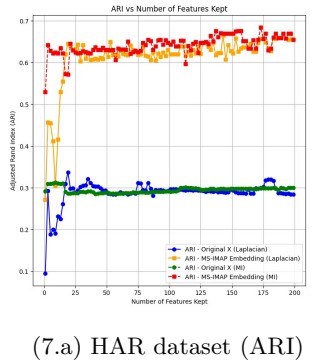

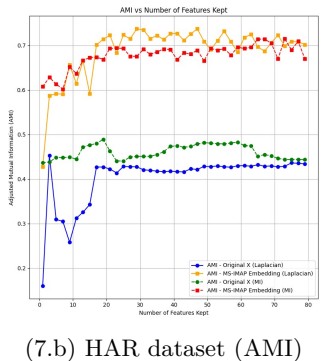

(7.a) HAR dataset (ARI)

(7.b) HAR dataset (AMI)

Figure 7: Clustering results (AMI) on the HAR datasets using a subset of features based on feature importance, comparing MS-IMAP embedding space and the input features.

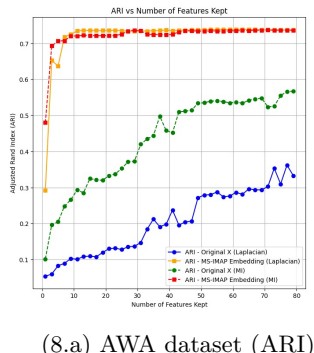

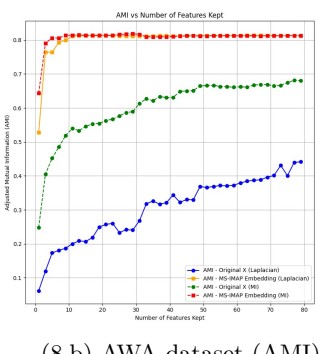

(8.a) AWA dataset (ARI)

(8.b) AWA dataset (AMI)

Figure 8: Clustering results ARI and AMI on the AWA dataset using a subset of features based on feature importance, comparing MS-IMAP embedding space and the input features.

On the HAR and AWA datasets, Figures 7, 8, show that, for both feature selection methods, clustering accuracy is consistently higher when using the MS-IMAP embedding than when using the original features—across a range of feature subset sizes and for both evaluation metrics. This improvement highlights the robustness of the MS-IMAP embedding, while maintaining interpretability through its one-to-one correspondence between embedding dimensions and original features.

### A.6 Dataset Details

**Two Moons.** The two moons dataset depicts two interleaving half-circles. We sampled $N = 600$ points and used a Gaussian noise level having standard deviation 0.12. An example is show in Figure 9. To produce these points we use Sci-kit Learn's `sklearn.datasets.make_moons`. In Figure 9, we show example clustering for each of the methods mentioned in the Experimental Results section of the main paper.

**Census.** From the UCI Machine Learning Repository Dua & Graff (2017), this dataset contains 14 features that are a mix of categorical, numerical, and binary. Such features include age, marital status, sex, etc. The goal is to predict whether a sample makes less than or equal to $50,000$, or strictly more. We use $32,561$ samples in our dataset.

**Lung Cancer.** The Zilionis dataset is widely used, and consists of single-cell RNA sequencing data. It has 306 features, and $48,969$ samples. The data has 20 classes corresponding to cell type. More can be found at Zilionis et al. (2019)

**Animals with Attributes.** The Animals with Attributes (AWA) dataset, contains 5,000 data points corresponding to 10 unseen classes, where the testing image features are obtained from the pre-trained ResNet architecture, with $D = 2,048$ dimensions, and the semantic features are provided with $D = 85$ dimensions. More information can be found in Section 4.1 of Xian et al. (2017).

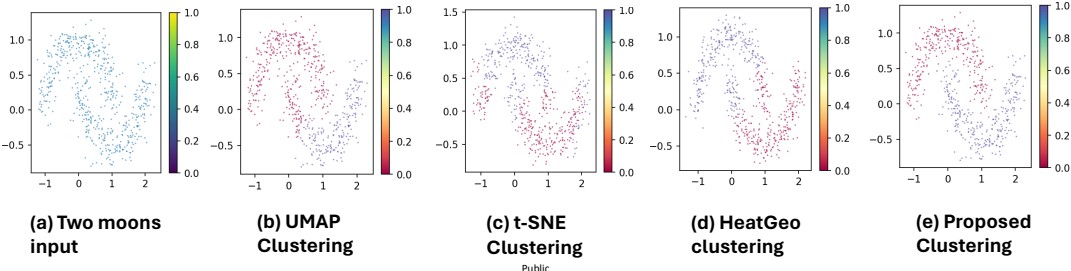

Figure 9: Clustering results using two moons

**Cora.** The Cora dataset has 2,708 scientific publications categorized into seven classes. The network has 5,429 links and each publication is described as a binary word vector indicating the presence or absence of a word. The dictionary consists of 1,433 words. In our experiments, we do not use the given graph, and only use the features. Sen et al. (2008); McCallum et al. (2000).

**HAR.** The Human Activity Recognition (HAR) dataset consists of sensor data collected from 30 subjects performing six different activities: walking, walking upstairs, walking downstairs, sitting, standing, and lying. The data was recorded using accelerometer and gyroscope sensors embedded in a smartphone worn on the waist. Each activity is represented as a feature vector of 561 attributes and 7352 samples, extracted from the raw sensor signals through signal processing techniques. In our experiments, we do not use the temporal dependencies and only utilize the extracted features.

## A.7 Experimental Results on two moons Synthetic Dataset

Here we show further experiments on more data, where we assess the robustness of our method using the two moons dataset, which is a 2D manifold depicting two interleaving half-circles. We sampled $N = 600$ points, and set the Gaussian noise standard deviation to 0.12. While spectral based methods such as UMAP are effective under relatively "modest" noise levels, their performance deteriorates in the presence of larger amounts of noise. As shown in Table 1 under the *Moons* column, our approach is robust and correctly clusters most points despite the large noise level, and competitive with the competing methods.

| Data | Two Moons dataset | |
|---|---|---|
| Method/Accuracy | ARI | AMI |
| Features | 0.24 | 0.18 |
| UMAP | 0.54 | 0.51 |
| t-SNE | 0.42 | 0.35 |
| ISOMAP | 0.36 | 0.3 |
| Scattering Geometric.T | 0.26 | 0.19 |
| Diffusion Maps | 0.25 | 0.19 |
| PHATE | 0.42 | 0.35 |
| HeatGeo | 0.54 | 0.52 |
| MS-IMAP | **0.89** | **0.87** |

Table 4: Comparison of clustering performance on the two moons dataset.

## A.8 Hyperparameter Tuning Details

We note that for UMAP, t-SNE, PHATE, PaCMAP, TriMap and HeatGeo, we hyperparameter tune the performance on each dataset, running the method five times for each hyperparameter configuration and taking the average. Then we report the best average score. Additional details regarding hyperparameter

are provided below.

We tune UMAP on each dataset in the Experimental Results section, by tuning over the parameter space in Table 5.

| Hyperparameter | Set of values |
| --- | --- |
| n_neighbors | 2, 10, 15, 20, 30, 50, 100 |
| min_dist | 0, 0.1, 0.5, 0.99 |
| n_components | 2, 3 |

Table 5: Space of parameters in which we tuned UMAP.

### A.9 TSNE Hyperparameter Tuning Details

We tune TSNE on each dataset in the Experimental Results section, by tuning over the parameter space in Table 6.

| Hyperparameter | Set of values |
| --- | --- |
| perplexity | 15, 30, 60, 200 |
| early_exaggeration | 12, 24 |
| n_components | 2, 3 |

Table 6: Space of parameters in which we tuned TSNE.

### A.10 PHATE Hyperparameter Tuning Details

We tune PHATE on each dataset in the Experimental Results section, by tuning over the parameter space in Table 7.

| Hyperparameter | Set of values |
| --- | --- |
| knn | 3, 5, 10 |
| decay (i.e. $\alpha$) | 6, 10, 14, 18, 40 |
| t | auto, 50, 100, 150, 200 |

Table 7: Space of parameters in which we tuned PHATE.

### A.11 HeatGeo Hyperparameter Tuning Details

We tune HeatGeo on each dataset in the Experimental Results section, by tuning over the parameter space in Table 8,

### A.12 Ablation Studies of Hyperparameters for MS-IMAP

Here we study what effect the hyperparameters – the number of nearest neighbors, and the number of filters – have on the clustering performance of MS-IMAP. In table 9, we demonstrate the effect of varying the number of neighbors, between 10, 15, and 20 neighbors. We see that the results are mostly the same, thus showing MS-IMAP is robust on real datasets.

We also study the effect of varying the number filters. In table 10, we also see similar results when using different filters, showing the stability of MS-IMAP.

| Hyperparameter | Set of values |
|---|---|
| knn | 5, 10, 15 |
| lap_type | normalized, combinatorial |
| harnack_regul | 0, 0.5, 1 |

Table 8: Space of parameters in which we tuned HeatGeo.

| Dataset | Census | | Zilionis | | AwA | |
|---|---|---|---|---|---|---|
| Number of Neighbors / Accuracy | ARI | AMI | ARI | AMI | ARI | AMI |
| 10 | 0.22 | 0.15 | 0.71 | 0.78 | 0.73 | 0.80 |
| 15 | 0.23 | 0.15 | 0.70 | 0.76 | 0.74 | 0.81 |
| 20 | 0.22 | 0.15 | 0.70 | 0.77 | 0.71 | 0.79 |

Table 9: Ablation study on MS-IMAP with varying the number of neighbors. The number of filters is kept at 5.

### A.13    Ablation Studies of Hyperparameters for t-SNE, Isomaps

We study the effect of varying the number of neighbors for the methods: t-SNE, Isomaps, and Diffusion Maps. For t-SNE this would be the perplexity hyperparameter, for Isomaps this would be the number of neighbors, and for diffusion maps, this would be the parameter that affects the width of the Gaussian kernel, i.e. $\exp(\cdot/\alpha)$.

In Table 11, we see choosing a smaller perplexity of 15 does worse than the perplexity of 30, 60 and 200.

In Table 12, we see a similar pattern to t-SNE, where choosing too low causes reduction in performance. But performance stabilizes around choosing the number of neighbors as 5-10+.

| Dataset | Census | | Zilionis | | AwA | |
|---|---|---|---|---|---|---|
| Number of Filters / Accuracy | ARI | AMI | ARI | AMI | ARI | AMI |
| 5 | 0.23 | 0.15 | 0.70 | 0.76 | 0.74 | 0.81 |
| 6 | 0.22 | 0.15 | 0.70 | 0.76 | 0.73 | 0.81 |
| 7 | 0.22 | 0.15 | 0.72 | 0.78 | 0.74 | 0.81 |

Table 10: Ablation study on MS-IMAP with encoding method 2, varying the number of filters. The number of neighbors is kept at 15.

| Dataset | Census | | Zilionis | | AwA | |
|---|---|---|---|---|---|---|
| Perplexity / Accuracy | ARI | AMI | ARI | AMI | ARI | AMI |
| 15 | 0.03 | 0.04 | 0.37 | 0.67 | 0.69 | 0.76 |
| 30 | 0.15 | 0.15 | 0.38 | 0.68 | 0.73 | 0.80 |
| 60 | 0.17 | 0.17 | 0.38 | 0.69 | 0.73 | 0.80 |
| 200 | 0.2 | 0.24 | 0.39 | 0.69 | 0.71 | 0.80 |

Table 11: Ablation study on t-SNE, varying the perplexity.

| Dataset | Census | | Zilionis | | AwA | |
|---|---|---|---|---|---|---|
| Neighbors / Accuracy | ARI | AMI | ARI | AMI | ARI | AMI |
| 2 | -0.05 | 0.02 | 0.41 | 0.55 | 0.43 | 0.58 |
| 5 | 0.18 | 0.09 | 0.44 | 0.56 | 0.52 | 0.60 |
| 10 | 0.18 | 0.09 | 0.44 | 0.57 | 0.51 | 0.62 |

Table 12: Ablation study on Isomap, varying the number of neighbors.

## A.14 Fast computation using Chebyshev polynomials

We provide additional details regarding the fast computation of SGW coefficients Hammond et al. (2011). Directly computing the SGW coefficients above requires calculating the entire eigensystem of the Laplacian, which is computationally intensive - $O(N^3)$ for $N$ points. Instead, Hammond et al. Hammond et al. (2011) suggested computing the SGW using a fast algorithm based on approximating the scaled generating kernels through low-order polynomials. The wavelet coefficients at each scale are then computed as a polynomial of $\mathbf{L}$ applied to the input data, using approximating polynomials given by truncated Chebyshev polynomials.

The Chebyshev polynomials $T_k(y)$ are computed using the recursive relations: $T_k(y) = 2yT_{k-1}(y) - T_{k-2}(y)$ for $k \geq 2$, where $T_0 = 1$ and $T_1 = y$.

The SGW coefficients are then approximated using wavelet and scaling function coefficients as follows:

$$\psi_f(s_j, i) \sim \left( \frac{1}{2}c_{j,0}f + \sum_{k=1}^{K} c_{j,k}\bar{T}_{j,k}(\mathbf{L})f \right)_i, \tag{27}$$

where $c_{j,k}, j > 0$ are the Chebyshev coefficients and $\bar{T}_{j,k}$ are the shifted Chebyshev polynomials $\bar{T}_k(x) = T_k\left(\frac{x-a}{a}\right)$ for $x \in [0, \lambda_{\max}]$, where $x = a(y+1)$, $a = \lambda_{\max}$. The scaling function coefficients, which are corresponding to a low-pass filter operation, are approximated in a similar way using Chebyshev polynomials. Note that the scaling kernel function is a low pass filter $h$ satisfying $h(0) > 0$ and $h(x) \to 0$ when $x \to \infty$. If the graph is sparse, we obtain a fast computation of the matrix-vector multiplication $\bar{T}_{j,k}(\mathbf{L})f$, where the computational complexity scales linearly with the number of points, resulting in a complexity of $O(N)$ for an input signal $f \in \mathbb{R}^N$. The SGWs efficiently map an input graph signal (a vector of dimension $N$) to $NK$ scaling and wavelet coefficients.

## A.15 Runtime and Computational Complexity

We have performed experimental runtime studies on empirical datasets. The execution time of our method with Python code implementation with 32 cores Intel Xeon 8259CL running at 2.50Ghz and 256GB of RAM on the Cancer QC data-set of 48,969 samples and 306 features took 27 mins, on Cora 5 min, on the AWA dataset 1.35 mins and on Census 7 min. In fields such as finance and genomics/biology, where typical dataset sizes range from under 100,000 to several million data points, analysis is often conducted offline once the graph embedding is computed. Runtimes of up to a few hours are generally acceptable and do not present

a significant challenge, making the method well-suited for real-world applications. The computational complexity of MS-IMAP is of $O(ND\log(N))$ for construction of the multi-scale representations which includes the $k$ nearest neighbor graph using k-d tree, the SGW transform which is $O(N)$ for each dimension of the manifold for sparse graphs. The optimization stage has a complexity which scale with the number of edges in the graph, which has a complexity of $O(kDN)$. Note that our approach generally requires the embedding dimensionality to be at least equal to the input dimensionality of the features, which adds computational complexity.

