# OpenReview forum: "MS-IMAP - A Multi-Scale Graph Embedding Approach for Interpretable Manifold Learning"
_TMLR — Accepted by TMLR_

### Review · Reviewer_6YrC · 2025-09-06

**Summary Of Contributions:**

1. This paper introduces a framework for multi-scale graph representation using a 3D tensor and contrastive learning.
2. This paper proposes a manifold-based interpretation method for feature importance.
3. This paper uses spectral graph wavelets to capture both global and local graph structures.

**Audience:**

Yes

**Audience Explanation:**

This paper introduces MS-IMAP, a graph embedding framework that addresses the common interpretability problem in manifold learning by combining spectral graph wavelets and contrastive learning. The core contribution is its ability to establish a one-to-one correspondence between embedding dimensions and original features, allowing for direct assessment of feature importance. Additionally, MS-IMAP uses spectral graph wavelets to capture both global and local graph structures, generating more expressive embeddings, while an efficient 3D tensor optimization method enhances its computational efficiency.

**Broader Impact Concerns:**

I have no concern.

**Claims And Evidence:**

Yes

**Claims Explanation:**

The paper provides extensive experimental validation on multiple public datasets to support its claims. It compares the performance of MS-IMAP against several baselines. The results show that MS-IMAP consistently outperforms the baselines, with the most significant improvements noted in feature importance evaluation. It also includes a theoretical justification for its approach, particularly for the use of spectral graph wavelets over traditional Laplacian operators.

**Requested Changes:**

The paper could be improved by revising its language, which at times reads like it was generated by a large language model and lacks depth. The authors should also pay attention to small formatting details, such as correctly punctuating inline formulas as part of a sentence.

Regarding the common challenge of selecting the number of dimensions for spectral clustering without prior information, the authors could strengthen their work by discussing and applying well-established model selection techniques. Promising methods include the eigengap heuristic, which finds the largest gap between consecutive eigenvalues of the graph Laplacian, and the silhouette score or Calinski-Harabasz index, both of which evaluate cluster quality for a range of k values. Incorporating an experiment using these methods would demonstrate a more robust and practical approach to their framework.

---

> ### Author Response · Authors · 2025-12-19
> **Response to Reviewer 6YrC**
>
> Regarding Requested Changes: "The paper could be improved by revising its language..."
>
> Response: We appreciate the suggestion  and will revise the manuscript to improve clarity and wording.
>
> Regarding "the common challenge of selecting the number of dimensions for spectral clustering without prior information, the authors could strengthen their work by discussing and applying well-established model selection techniques. Promising methods include the eigengap heuristic, which finds the largest gap between consecutive eigenvalues of the graph Laplacian, and the silhouette score."
>
> Response: To further clarify the effect of feature selection on clustering structure, we report the internal clustering metric of the silhouette score on the selected feature subsets. This label-free measure does not use ground-truth labels and therefore does not replace our primary ARI/AMI evaluation, but it provides an additional unsupervised check that the selected subsets induce coherent clusters in the embedding space.
>
> The Figures in Section A.2, in the new manuscript show the silhouette score as a function of the number of selected features $k$ when Laplacian Score is applied either to the original features or to the MS-IMAP embedding on HAR, Zillion, and Cora datasets, respectively.
>
> In these experiments, for each dataset and each number of selected features $k$, we run $k$-means with several values of number of clusters $K_c$ and report the best silhouette score, which serves as a simple silhouette-based model selection over the number of clusters.
>
> For HAR and Zillion, the MS-IMAP embedding consistently achieves higher and more stable silhouette scores (around $0.6$–$0.7$) than the original features, whose scores steadily degrade as $k$ increases. For Cora, the original space exhibits a very high silhouette for the top one or two features but then rapidly collapses toward zero as more features are added, whereas the MS-IMAP embedding maintains a moderate but stable silhouette (around $0.3$) over a wide range of $k$. Overall, MS-IMAP–based feature selection trades a narrow peak in silhouette for more robust cluster quality once $k$ exceeds a few features. Regarding the question of selecting the number of clusters or embedding dimensions (e.g., via eigengap or silhouette-based model selection), we view this as an important but orthogonal problem and leave a systematic study of such strategies in combination with MS-IMAP to future work.

---

> > ### Comment · Reviewer_6YrC · 2026-01-10
> >
> > Thank you for the thorough rebuttal and the added analysis. The new silhouette-based internal clustering evaluation in Appendix A.2 is a helpful, practical complement to the ARI/AMI results, and it addresses my main concern about providing an unsupervised sanity check for model/feature selection.

---

### Review · Reviewer_1k5z · 2025-10-15

**Summary Of Contributions:**

# Summary

**MS-IMAP** applies Spectral Graph Wavelets with contrastive learning to produce interpretable embeddings that preserve a one-to-one mapping between input features and embedding dimensions. It encodes multi-scale graph structures with 3D tensors, optimizes using SGD and cross-entropy loss, and provides theoretical grounding through Paley–Wiener spaces.
Experiments on five datasets demonstrate strong clustering performance and enable feature importance analysis using Laplacian Score (LS) and Mutual Information (MI).

---

# Strengths

### Interpretability Contribution
MS-IMAP’s one-to-one feature mapping fills a real gap in manifold learning. Unlike UMAP, t-SNE, or Laplacian Eigenmaps, it allows direct tracing from embedding dimensions to input features, which is especially valuable in domains like finance and biology.

### Comprehensive Evaluation
The study includes thorough experiments with well-tuned baselines, diverse datasets across multiple domains, detailed ablation studies, and competitive results that are best or second-best on all benchmarks.

### Dual Feature Importance Methods
Using both LS (unsupervised smoothness) and MI (task-aware discriminative power) provides complementary insights into feature relevance, as clearly discussed in Section 4.3.

---

# Weaknesses

### Unfair Dimensionality Comparison
MS-IMAP retains the full input dimension \(D\) (from 85 to 2,048), while baselines are reduced to two or three dimensions. This gives an inherent advantage in clustering performance and contradicts the manifold learning assumption that data lies in lower-dimensional spaces.

### Circular Feature Importance Validation
Pseudo-labels are created using k-means on MS-IMAP embeddings, and MI is then computed between these embeddings and their own derived labels. This forms a circular validation process, meaning improvements mainly reflect self-consistency rather than independent verification.

**Audience:**

Yes

**Audience Explanation:**

The paper addresses interpretable manifold learning, a timely problem given increasing demand for explainable ML. The novel one-to-one feature correspondence, rigorous theoretical foundation connecting harmonic analysis to graph learning, multi-scale representation approach, and results on practical datasets across multiple domains (biology, finance, vision) would appeal to interpretable ML researchers, graph learning community, and practitioners in high-stakes applications

**Claims And Evidence:**

Yes

**Claims Explanation:**

The theoretical claims are rigorously proven (Theorems 2-3 with full proofs), clustering performance is demonstrated across 5 diverse datasets with proper statistical reporting and extensive baseline comparisons (9 methods with hyperparameter tuning), and the feature importance advantage is consistently shown in Figures 2-3, 5-6.

**Requested Changes:**

Please refer to the weakness.

---

> ### Author Response · Authors · 2025-12-19
> **Response to Reviewer 1k5z**
>
> Regarding "Unfair Dimensionality Comparison: MS-IMAP retains the full input dimension":
>
> We refer to the general response to Dimensionality Comparison in the beginning of the rebuttal and the new section A.1 in the manuscript.
>
> Regarding "Circular Feature Importance Validation: “Pseudo-labels are created using k-means on MS-IMAP embeddings, and MI is then computed between these embeddings and their own derived labels. This forms a circular validation process...”
>
> We agree that MI computed with pseudo-labels derived from MS-IMAP embeddings is a form of self-supervised evaluation and is not an independent ground-truth check by itself. In our experiments, however, this pseudo-label–based feature ranking is not used as the final validation criterion: the quality of the resulting feature subsets is evaluated using external clustering metrics (ARI / AMI) with respect to the true labels.
> The role of the pseudo-label MI is to suggest an alternative to LS where self-supervised feature selection is more effective. We observe that MI-based feature selection performed in the MS-IMAP embedding space consistently yields better clustering performance than applying the same pseudo-label–based selection in the raw input space (Fig. 2–3, Sec. 5.1). This suggests that MS-IMAP produces an embedding in which the cluster structure is more semantically aligned with the intrinsic graph geometry. In addition, we report feature-selection results based on Laplacian Score (LS), which depends only on the intrinsic graph structure and does not rely on pseudo-labels. LS-based selection in the MS-IMAP embedding also improves clustering performance, which supports the robustness of our feature-importance claims beyond the specific choice of MI with pseudo-labels. To strengthen our argument, we also expand the feature-selection analysis in new Sec. A.2 in the Appendix.

---

### Review · Reviewer_rd3L · 2025-12-05

**Summary Of Contributions:**

The paper introduces the method MS-IMAP, a new representation learning method. It uses spectral wavelets at multiple scales and learns an embedding via contrastive learning. Notably, the dimensionality of the learned embedding is the same as that of the input data. This one-to-one correspondence between embedding and original features allows the definitions of importance scores for the original features. They evaluate their method quantitatively by the clustering performance on various datasets and compare to dimensionality reduction / visualization methods and a graph scattering approach.

**Strengths:**

S1 The one-to-one correspondence between original and embedding features allows to compute meaningful feature importance scores.
S2 The method explicitly incorporates several frequency scales and can thus capture both broad and fine data structure.
S3: The authors optimize hyperparameters for each competing method.

**Weaknesses:**
*Major:*
W1 Comparison mostly to visualization methods

MS-IMAP is not a dimensionality reduction or visualization method. Indeed, it maintains the same number of dimensions as the input had. Therefore, it is not meaningful to almost exclusively compare to dimensionality reduction / visualization methods, which only have 2-3 dimensions, when MS-IMAP has hundreds to thousands. If anything, the often slim margins by which MS-IMAP outperforms methods with order of magnitude less capacity cast doubt on the effectiveness of MS-IMAP. As the authors mention in Sec 5.2. the low embedding dimension of visualization methods leads to reduced expressiveness. As a result, I do not find the results in section 5.2 meaningful even if they were to show a clear performance increase for MS-IMAP. However, in many columns of Table 1 MS-IMAP gets outperformed by or draws with a method of significantly less capacity.

I think a much more meaningful comparison would be to compare to unsupervised deep representations, e.g., via auto-encoders or self-supervised graph neural networks (keeping one or two dimensionality reduction methods in the comparison is fine, I think). Several of the dimension reduction methods allow embedding to higher-dimensional spaces: E.g. Diffusion Maps approximate the diffusion distance which is exactly represented by $N$-dimensional embeddings. Similarly, PHATE defines a graph distance, the potential distance, which is realized as Euclidean distance between points in $N$ dimensions, and then runs MDS to reduce the dimensionality. One could simply omit the MDS step, or use a much higher embedding dimension. One could run UMAP (with random initialization to avoid long Laplacian Eigenmaps computation) to much higher dimensions. Possibly, the methods from Gao et al 2018  2019 and Sun et al 2009 would also be relevant competitors.

W2 Isolation of different features
I appreciate that during the optimization, only values for the same feature get combined. However, statements such as

>  each embedding coordinate is updated solely based on the SGW coefficients of its corresponding input feature

in the Discussion are too strong because the structural term $v_{ij}^{\psi_x(s_k, :, :)}$ depends on all features jointly. In a similar vein, the graph structure itself depends on all features jointly.

*Minor:*
W3 Expand on feature importance by using MS-IMAP selected features in other methods
As I understand figures 2 and 3, they show the performance on a subset of the original features. This subset is selected by importance scores either computed on the original or the MS-IMAP features. For the MS-IMAP results, the selected features are again fed into MS-IMAP. I think it would be good to highlight how useful these features are more generally. For instance, one could compute the clustering performance of the MS-IMAP features without additional application of MS-IMAP or as input to some of the other methods considered in the paper (graph scattering, Diffusion Maps, t-SNE, UMAP). In addition, it would be great if the authors could comment qualitatively on the selected features. Which features get selected? Is it plausible that these carry semantic relevant to the clustering task?

W4 Section 6 hard to understand and disconnected
I am not an expect on spectral graph wavelets and did not check the correctness of the theoretical claim Theorem 3. However, I can say that a non-technical, high-level, natural language explanation of why Theorem 3 exhibits desirable properties of MS-IMAP is needed. What do $\Lambda$-sets mean for representation properties of an operator. Is a small / low $\Lambda$-set desirable? If so, why? Overall this section feels somewhat disconnected from the rest of the paper.


W5 Missing competitors
The authors describe Gao et al. 2018, Gama et al. (2019), Sun et al. (2009) as related methods, because they compute higher-dimensional graph features. However, they only compare against Gama et al.'s method. Why are the other two not relevant for the empirical comparison?

W6 No code was submitted.

**Typos:**
- Full stops should be after the equations, not on the next line on page 4.
- UMAP's ARI score on the Cora dataset is as good as MS-IMAP's in the shown digits, but not bold in Table 1
- TriMap's and PaCMAP's ARI scores on the AWA dataset are equal to or better than the bold ones in that column of Table 1.
- Appendix A.1 ends in the middle of a sentence (that was not capitalized)
- "Multi-Scale UMAP" --> "Mulit-Scale IMAP" on page 21.
- UMAP uses an exponential kernel, not a Gaussian kernel (Sec 2.1)

**Questions:**

Q1 What does the acronym MS-IMAP stand for?

Q2: What doe the following notations mean?

$\hat{f}$ in Eq (1)

$\tilde{\psi}_x$ in Eq (2)

$\psi_{x_i}$ after Eq (3) (is it the same as $\psi_x(:, :, i)$)?

Q3: What is meant by weighing each scale by its energy concentration? Where can I find this in the equations describing the model?

Q4: Are there any hyperparameters of MS-IMAP that get optimized per dataset? Which number of neighbor was used for the graph construction? Which wavelet scales were used?

**Audience:**

Yes

**Audience Explanation:**

The feature selection approach seems promising and interpretability is a pressing issue. The spectral graph wavelet approach is conceptually interesting, but need to be validated empirically in a more convincing way.

**Claims And Evidence:**

No

**Claims Explanation:**

I think that several claims in the paper have problematic supporting evidence. The biggest issue to me is that MS-IMAP, which does not reduce the dimensionality at all, is almost exclusively compared to methods that reduce the dimensionality by several orders of magnitude to 2 or 3 dimensions (W1, W5). This is an apples-and-oranges comparison. Second, I think that there is more interaction between the different features (through the graph and the structural term), than the authors claim (W2). The theoretical section could be explained more accessibly and connected more closely to the proposed method (W4).

**Requested Changes:**

C1: Include several higher capacity competitors (e.g. Gao et al., (graph) autoencoders, deep (graph) SSL methods, high-dimensional UMAP, DiffMaps, PHATE variants) to determine if MS-IMAP outperforms models that are not handicapped by extremely low-dimensional representations (W1, W5).

C2: Adapt the description of feature isolation (W2).

C3: Expand the feature selection section (W3).

C4: Explain the relevance of the theoretical section more clearly (W4).

C5: Address typos and questions above.

---

> ### Author Response · Authors · 2025-12-19
> **Response to Reviewer rd3L**
>
> W1: See our general response and new Sec. A.1 in the appendix.
>
> W2: We agree that the structural term $v_{ij}^{\psi_x}$ and the graph construction depend jointly on all features, but we emphasize that this influence is indirect. Each embedding coordinate for a sample is computed from SGW coefficients that are derived from the graph signal of a single original feature, so there is a direct feature-to-coordinate mapping, with joint structure entering only through the shared graph. The graph structure is the same for all embedding coordinates, while the graph signal (one feature at a time) changes across coordinates. This is consistent with many interpretability frameworks, where the contribution of a feature is quantified in the presence of all others rather than assuming independence. In our manifold-learning setting, we explicitly do not assume feature independence; instead, MS-IMAP maintains a one-to-one correspondence between features and embedding coordinates while leveraging joint structure on the graph. We clarify this distinction and soften the wording in the revised Discussion.
>
> W3: Sec. 5.1 evaluates unsupervised feature selection within the MS-IMAP framework. We select feature subsets using MI with pseudo-labels (based on running $k$-means either in the embedding space or in the input space) and Laplacian Score (LS), and then test these subsets directly on clustering tasks (ARI/AMI) to compare subsets derived from the MS-IMAP embedding against subsets selected from the raw input features. Using MS-IMAP–selected subsets as inputs to other manifold-learning or graph-embedding methods (e.g., scattering, Diffusion Maps, t-SNE) is an interesting direction for exploiting the regularized structure learned by MS-IMAP, but is secondary to our main objective, which is to construct a transparent and robust embedding space with a built-in feature–embedding correspondence, rather than stacking multiple black-box embeddings. To strengthen our argument, we expand the feature-selection analysis in new Sec. A.2.
>
> W4: We add details in Sec. 6 explaining what $\Lambda$-sets measure, why smaller values are better, and how this relates to sample efficiency and stability, and explicitly connect Theorem 4 back to MS-IMAP by stating that the improved sampling bound supports the use of SGW as the core operator in our embedding framework.
>
> W5: We cited Gao et al. (2018) and Sun et al. (2009) as conceptually related in their use of higher-dimensional graph representations, but they target different problem settings than MS-IMAP and are not straightforward baselines for our experiments. Gao et al. (2018) use graph scattering transforms primarily for graph classification on networks, whereas our setting is manifold learning and clustering on a single graph constructed from a point cloud with feature-wise interpretability constraints. Sun et al. (2009) focus on a shape correspondence problem, which differs from our unsupervised feature-aligned embedding and clustering setting. By contrast, Gama et al. (2019) and the more recent HeatGeo method operate in a setting closer to ours and can be more naturally adapted to our benchmark protocol, so we include them as empirical baselines.
>
> W6: Due to institutional proprietary constraints, we are unable to release the code implementation, but we have clarified the key implementation details.
>
> Q1: MS-IMAP stands for Multi-Scale Interpretable Manifold Approximation and Projection.
>
> Q2: We clarified and streamlined the notation throughout the manuscript.
>
> Q3: In our setting, each spectral graph wavelet scale $s_k$ corresponds to a different frequency (spectral) band of the graph Laplacian: lower scales emphasize low frequencies (smooth variations on the graph), while higher scales emphasize higher frequencies. The SGW filters $g(s_{k} \lambda_l)$ in Eq. (1) determine how much of the signal’s energy is captured at each scale as a function of the Laplacian eigenvalues $\lambda_l$. Different scales therefore capture different portions of the spectrum, and this distribution depends on the graph structure through its eigenvalues. Each scale focuses on a particular spectral band, and the amount of signal it captures (what we referred to as its “energy”) is determined jointly by the SGW filter and the Laplacian spectrum. The resulting “weighting” across scales is implicit in the SGW construction via $g(s_{k} \lambda_l)$ and the spectrum of $\mathbf{L}$, rather than an additional per-scale parameter in the loss. We clarify this point in the revised manuscript.
>
> Q4: A detailed report of hyperparameter tests is provided in the Appendix. In all experiments, we use $k = 15$ nearest neighbors for the kNN graph and $K = 5$ SGW scales with the Mexican hat filter.  We note this contrasts with other methods (not ours), where we give them the benefit of hyperparameter tuning on each dataset.

---

> > ### Comment · Reviewer_rd3L · 2025-12-21
> >
> > Many thanks for the reply and the revised manuscript! Some of my concerns have been addressed (W4, W5). Other issues remain (partly):
> >
> > W1: Many thanks for adding appendix A1 with competitors using higher-dimensional embeddings. This addresses much of my concern, but I would also appreciate
> > - to see the results on the remaining datasets
> > - correct highlighting of the best method in Table 3 (UMAP outperforms MS-IMAP on Cora and draws on Zilionis in AMI)
> > - a discussion and cross-reference of appendix A1 in the main paper, so that readers do not overlook this important comparison (e.g. in the results section 5.2 or the discussion)
> >
> > W2: I agree that the joint influence of the features through the graph and the structural term seems indirect. But there is no quantitative evaluation of the effect, so I remain unconvinced that features are mainly independent. E.g. one could compute a separate graph per input feature and compute structural terms per dimension, to ensure that different features truly cannot influence each other. I also do not see a clarification or softening of this fact in the discussion section of the revision. In fact, the sentence
> >
> > >  each embedding coordinate is updated solely based on the SGW coefficients of its corresponding input feature.
> >
> > is just as in the original submission. This remains a key issue. In case I overlooked the discussion, please point me to the corresponding line number.
> >
> > W3: The rebuttal and the reordering of the results sections puts more focus on the features importance than the clustering section, which I think is good as it reflects the fact that the clustering performance does not go beyond state of the art. However, given this refocusing on interpretability and feature selection, the amount of experiments in this section seems a bit thin. Currently, there are only two competitors (MI and LS on the original features) and only three datasets are considered (Cora, Zilionis in main paper, HAR in appendix). There are many other feature selection methods, e.g. the typical highly-variable gene selection pipelines for scRNA-seq data [a]. If the key contribution of MS-IMAP is good feature selection, I recommend comparing to more methods on all considered datasets.
> >
> > References:
> >
> > [a] Luecken, M. D., & Theis, F. J. (2019). Current best practices in single‐cell RNA‐seq analysis: a tutorial. Molecular systems biology, 15(6), e8746.

---

> > > ### Author Response · Authors · 2025-12-25
> > > **Response to Reviewer rd3L**
> > >
> > > Regarding W1
> > >
> > > Response: We thank the reviewer for these comments. As requested, we have
> > > (i) added results on the remaining datasets (see Appendices A.1 and A.2),
> > > (ii) corrected the highlighting of the best methods in Table 3, and
> > > (iii) added explicit cross-references to Appendix A.1 in the main text
> > > (Results / Discussion sections). These changes are included in the revised manuscript uploaded to the site.
> > >
> > > Response to W2 (feature isolation / “solely based on”).
> > >
> > > We apologize for the confusion. In the previous revision, the sentence in the Discussion was not fully updated, even though additional clarification was added in Sec. 3.2. In the revised manuscript we now explicitly modify this claim. The wording in the Discussion now reads:
> > >
> > >  “Importantly, although SGW captures information across the graph, each embedding coordinate remains feature-aligned (one per input feature) while its update incorporates joint structural coupling through the shared graph and SGW operator.”
> > >
> > > We also expanded Sec. 3.2 to make this mechanism more explicit. In particular:
> > >
> > > (i) each embedding coordinate remains feature-aligned in the sense that its update uses SGW coefficients (\psi_x(s_k,d,:)) associated with a single original feature (d); we do not mix SGW channels from different features within a single coordinate;
> > >
> > > (ii) at the same time, the structural term (v_{ij}^{\psi_x}) and the underlying graph Laplacian depend on all features jointly, so different coordinates are still coupled through the shared graph and the contrastive loss. Our interpretability guarantee is therefore a one-to-one correspondence between features and embedding coordinates, not statistical independence between features in the learned representation.
> > >
> > > A stronger notion of isolation such as constructing a separate graph for each input feature and defining a structural term per dimension would prevent different features from influencing one another via the graph. However, this would amount to assuming an independent geometry for each feature and would discard the joint manifold structure that motivates MS-IMAP in the first place, while also being substantially more expensive computationally. We therefore do not aim to enforce such full independence and have adjusted the manuscript wording to make clear that the embedding is feature-aligned while still incorporating joint structural coupling through the shared graph.
> > >
> > > Regarding W3 (focus on feature importance / number of baselines and datasets).
> > >
> > > Response: As clarified in the main text, the main contribution of our work is a novel, interpretable graph embedding mechanism that addresses a long-standing challenge in interpretable machine learning. Clustering and unsupervised feature selection are presented as proof-of-concept applications of this embedding, demonstrating that the feature-aligned correspondence between input features and embedding coordinates can be exploited in practice for downstream tasks (e.g., via MI- and LS-based ranking).
> > >
> > > MI and LS were selected as baselines because they are generic, unsupervised, applicable across all datasets we consider, and they probe complementary aspects of the representation (pseudo-label alignment and graph smoothness). Our goal in Sec. 5.1 is therefore not to exhaustively benchmark all feature-selection algorithms, but to show that standard criteria become more effective when applied in the MS-IMAP embedding space than in the raw feature space.
> > >
> > > We agree that incorporating additional, domain-specific pipelines such as highly variable gene-selection schemes tailored to scRNA-seq data could further enrich the empirical picture on Zilionis-like datasets. However, many of these methods are specialized to particular data modalities, and a comprehensive comparison across all such pipelines is better viewed as an extension rather than the focus of this initial study.
> > >
> > > In addition, we strengthened the feature-selection evaluation by reporting label-free clustering diagnostics (silhouette scores) on multiple datasets in Appendix A.2, which further supports that MS-IMAP based feature subsets improve intrinsic cluster structure. More generally, using MS-IMAP as a basis for Shapley-value attributions or other explainability tools is a promising direction for future work that builds on, but goes beyond, the scope of the current paper.

---

> > > > ### Comment · Reviewer_rd3L · 2026-01-07
> > > >
> > > > Many thanks for the additional revision. Weakness W2 is now resolved from my side. While the silhouette score results are somewhat mixed (high performance for on Cora for the first few selected original features) W3 is also somewhat addressed. I also thank the authors for cross-referencing Appendix A1 and including more datasets. However, I am surprised that some numbers in Table 3 are lower than the corresponding values in Table 1 (e.g. UMAP in both metrics on Census, PHATE in both metrics on Zilionis). Why is that?

---

> > > > > ### Author Response · Authors · 2026-01-09
> > > > > **Response to Reviewer rd3L**
> > > > >
> > > > > The discrepancy between Table 1 and Table 3 comes from differences in the evaluation protocol rather than from MS-IMAP itself.
> > > > >
> > > > > In Table 1, each baseline (including UMAP and PHATE) was run with its own hyperparameter sweep tuned for the standard low-dimensional use case (e.g., recommended configurations at 2–3 dimensions), and we reported the best result from that sweep.
> > > > >
> > > > > In Table 3, the goal is specifically to study the effect of embedding dimensionality. To keep this experiment tractable, we fixed the remaining hyperparameters across the different values of
> > > > >
> > > > > $d \in {2,5,10,20,50,D}$ rather than performing a full hyperparameter search for every method, $d$ and dataset combination. As a consequence, the exact configuration that achieved the best scores in Table 1 is not guaranteed to appear among the Table 3 runs, so the “best over $d$” values in Table 3 can be slightly lower than the Table 1 numbers for some methods and datasets.
> > > > >
> > > > > Thus Table 3 show that under a consistent and reasonably tuned setup across dimensions, increasing the embedding dimension beyond 2–3 does not systematically improve the performance of these baselines, and that MS-IMAP remains competitive in this higher-capacity regime.

---

### Author Response · Authors · 2025-12-19
**General Response to Reviewers on Key Points**

We thank the reviewers for their thoughtful feedback and constructive suggestions, many of which we have incorporated into the revised manuscript. We begin by addressing common key points and clarifications.

Dimensionality comparison. We clarify (e.g., in Sec. 5.2 and in new Sec. A.1 in the Appendix) that MS-IMAP is not a dimensionality reduction or visualization method; it is an interpretable graph embedding that preserves a one-to-one correspondence between input features and embedding coordinates by design, and thus retains the original feature dimensionality rather than compressing to 2–3 dimensions. Section 5.2 is meant to situate MS-IMAP with respect to standard manifold learning and visualization baselines and to illustrate that, even under this feature–embedding alignment constraint, its performance remains comparable to these methods. We do not claim that higher dimensionality alone explains MS-IMAP’s performance, and we agree that this point should be made more explicit. Using higher dimensionality alone often does not guarantee improved performance, as illustrated in our comparison with graph scattering transforms and in the additional results presented in the rebuttal.

To address the fairness concern, in the revision we add comparisons (see Sec. A.1 in the Appendix) to higher-capacity unsupervised representation methods (e.g., autoencoder) and to higher-dimensional variants of manifold learning methods using multiple choices of embedding dimensionality. This directly tests whether MS-IMAP’s benefits persist when competing methods are not constrained to 2–3 dimensions and makes the role of representational capacity more transparent. We specifically address the reviewers' concerns regarding higher-capacity baselines and the effect of using a larger number of embedding dimensions for competing methods as follows.

(1) Unsupervised deep representations. As requested, we add a Graph Auto-Encoder (GAE) baseline following Kipf and Welling (2016), implemented via PyTorch Geometric’s GAE class, with a 2-layer GCN encoder (Kipf and Welling, 2017) and an inner-product decoder trained in a self-supervised fashion to reconstruct the $k$NN adjacency matrix built from the input features. We tune the embedding dimensionality over $d \in \{2, 5, 10, 20, 50, D\}$ where $D$ is the input dimensionality, and report in Table~\ref{tab:dimensionality_experiment} the best clustering performance (ARI/AMI) achieved by GAE over these values of $d$ for each dataset. (2) High-dimensional manifold learning baselines. We likewise re-run the manifold learning methods UMAP, HeatGeo, and PHATE with the same set of output dimensions, and again report the best performance over $d$ for each method and dataset.

The experimental results show that, in most cases, increasing the embedding dimension beyond the standard 2–3 dimensions does not yield substantial performance gains for these methods. Across datasets, MS-IMAP remains competitive with state-of-the-art baselines, typically achieving the first- or second-best ARI/AMI. Regarding GAE in particular, we observe that even with careful tuning of $d$ its clustering performance on these tabular $k$NN-graph datasets is comparable to or below that of simpler manifold-learning baselines. This is consistent with prior observations that unsupervised deep graph encoders can struggle to produce effective embeddings for clustering when the underlying graphs are irregular or constructed from tabular features rather than arising from a natural network structure. Our primary motivation for retaining input dimensionality is interpretability: most nonlinear embeddings sacrifice a direct, feature-wise correspondence that is central to the feature-importance results in Sec. 4–5.1. For additional details we refer to new Sec. A.1 in the Appendix.

Feature selection: To address requests regarding feature selection evaluation, we clarify the effect of feature selection on clustering structure using the internal clustering metric of the silhouette score on the selected feature subsets (new Sec. A.2 in the Appendix). For each dataset and each number of selected features $k$, we run $k$-means with several choices of the number of clusters $K_c$ and report the best silhouette score over these values of $K_c$, providing a simple silhouette-based model selection over $K_c$. This label-free measure does not use ground-truth labels and therefore does not replace our primary ARI/AMI evaluation, but it provides an additional unsupervised check that the selected subsets induce coherent clusters in the embedding space. MS-IMAP embedding consistently achieves higher and more stable silhouette scores over a wide range of $k$. Overall, MS-IMAP–based feature selection trades a narrow peak in silhouette for more robust cluster quality once $k$ exceeds a few features.

---

### Comment · Action_Editor_wzUF · 2026-03-31
**Camera-ready margins**

Dear authors,

Please make sure that the camera-ready version conforms to the TMLR style file (compare to the example paper for the look).

It appears that the margins are quite a bit wider than intended. Make sure you don't have custom commands changing the margins.

AE

---

### Decision · Action_Editor_wzUF · 2026-02-12

**Recommendation:** Accept with minor revision

**Additional Comments:**

Based on the reviews, the rebuttals, and the revisions, I recommend accepting this paper with two minor changes:
* Include the explanation of the discrepancies between Table 1 and Table 3 given to Reviewer rd3L as a note in Appendix A.1.
* Use the official TMLR style file—there appears to be a formatting inconsistency introduced. Compare to e.g., https://openreview.net/pdf?id=HaaAY4ZXPa

**Audience:**

Yes

**Audience Explanation:**

Yes, this was pointed out by multiple reviewers. The topic is timely and of interest to several communities in the TMLR audience.

**Claims And Evidence:**

Yes

**Claims Explanation:**

Reviewers found that the paper meets the bar for evidence to support its claims. One of the reviewers opposed the paper's acceptance in their recommendation and pointed out remaining question marks surrounding its claims. However, this recommendation was made before the final revision and before the final exchange between the reviewers and the paper's authors. The action editor confirmed with the reviewer separately that these issues have now been addressed and that the paper's claims are sufficiently supported for them to "lean towards acceptance".